# Integrated wind farm layout and control optimization

Mads M. Pedersen[1], Gunner Chr. Larsen[1]

[1]Wind Energy Department, Technical University of Denmark, Frederiksborgvej 399, DK-4000 Roskilde, Denmark

*Correspondence to*: Mads M. Pedersen (mmpe@dtu.dk)

**Abstract.** The objective of this paper is to investigate the joint optimization of wind farm layout and wind farm control in terms of power production. A successful fulfilment of this goal requires: 1) an accurate and fast flow model; 2) selection of the minimum set of design parameters that rules/governs the problem; and 3) selection of an optimization algorithm with good scaling properties.

For control of the individual wind farm turbines with the aim of wind farm production optimization, the two most obvious strategies are wake steering based on active wind turbine yaw control and wind turbine derating. The present investigation is limited to wind turbine derating.

A high-speed linearized CFD RANS solver models the flow field and the crucial wind turbine wake interactions inside the wind farm. The actuator disk method is used to model the wind turbines, and utilizing an aerodynamic model, the design space of the optimization problem is reduced to only three variables per turbine – two geometric and one carefully selected variable specifying the individual wind turbine derating setting for each mean wind speed and direction.

The full design space spanned by these ($2N+N_d N_s N$) parameters, where $N$ is the number of wind farm turbines, $N_d$ is the number of direction bins, and $N_s$ is the number of mean wind speed bins. This design space is decomposed in two subsets, which in turn define a nested set of optimization problems to achieve a significantly faster optimization procedure compared to a direct optimization based on the full design space. Following a simplistic sanity check of the platform functionality regarding wind farm layout and control optimization, the capability of the developed optimization platform is demonstrated on a Swedish offshore wind farm. For this particular wind farm, the analysis demonstrates that the expected annual energy production can be increased by 4% by integrating the wind farm control in the design of the wind farm layout, which is 1.2% higher than what is achieved by optimizing the layout only.

## 1. Introduction

The large-scale global deployment of wind energy is highly dependent on the cost of energy (COE), i.e. the profit of a wind power plant (WPP) over its lifetime as seen from an investor's perspective. Lowering the COE was previously addressed with the Topfarm WPP layout optimization platform (Réthoré et al., 2013; Larsen and Réthoré, 2013). The platform is used to design a WPP with minimal COE, for a given number of a predefined wind turbine (WT) type and an allowable area with an a priori known wind climate. Hence, it determines the optimal balance between WPP power production revenue on the one hand, and, on the other hand, all relevant expenses. The considered expenses include: WPP variable capital costs (i.e. capital costs that depend on the WPP layout); WPP operation and maintenance (O&M) costs, and cost of fatigue degradation of the individual components of all WTs in the WPP. The basic functionality of the Topfarm platform was later extended by also including the number of WPP WTs as a design variable, and the performance of surrogate models, needed to facilitate the used optimization algorithm, was moreover improved by Mahulja et al. (2018). Because WT loading is included, the WPP WTs must be modelled as aero-elastic models (including individual WT control), and the inflow conditions to these are tightly coupled to the complex non-stationary wake affected WPP flow field. Performing individual WT aero-elastic simulations for all the considered ambient wind speeds and wind directions in each layout configuration iteration is extremely costly in terms of computational efforts. Therefore, surrogate models are needed to link ambient WPP inflow conditions, WT location within the WPP, and WT response in terms of power production and (fatigue) loading.

However, WPP control aspects were not considered in the aforementioned WPP layout optimization platform. Fathy et al., (2001) presents a pure theoretical analysis of coupled design and control of general physical systems. It was found that conventional sequential optimization processes are not guaranteed to find system-optimal designs. In this theoretical framework, a coupling term is introduced, which reflects the influence of plant dynamics/control on plant design. The necessary conditions for the combined plant design and controller optimality was investigated, and it was concluded that this term depends strictly on the gradients of the couplings with respect to the plant design variables, which is also intuitively clear. Therefore, for weak/no coupling, i.e. neglectable coupling constraint gradients, the plant design and controller optimization problems become separable, and their sequential solution is equivalent to the combined optimum. In case of a strong coupling, only design methods that include this interaction explicitly can produce system-optimal designs contrary to the sequential approach. A priori, however, it is not possible to evaluate whether the coupling between system-design variables and system-control variables is weak or strong for a complicated physical system like a WPP.

Fleming et al. (2016) and Gebraad et. al (2017) study optimization of layout and active wake control in terms of WT yaw dictated wake deflection on a WPP with 60 WTs. In these studies, the wake effects are modelled with an augmented version of the N. O. Jensen model (Jensen 1984) extended with an engineering model for wake deflection as caused by WT yaw misalignment. Fleming et al. (2016) consider an inflow wind speed of 8 m s$^{-1}$, only, and report a power gain of 2.3% for the optimized layout; 7.6% for the optimized yaw control; and 8.5% for the integrated layout and yaw control optimization result. Finally, Fleming et al. (2016) compares the integrated result, which requires 6900 CPU hours, with a sequential approach, which can be performed in "*several hours by a single computer*". They find that the integrated result is around 0.5% better than the results originating from the sequential approach. Gebraad et. al (2017) perform a three-step optimization: first the AEP is increased by 1.5% by optimizing the layout considering one wind speed per wind direction, only; then the WT positions and the yaw angle are optimized, again based on one wind speed per wind direction, which increases the AEP to 5.2% above the base line. Finally, the WT yaw angles are optimized for all relevant wind speeds raising the AEP to 5.3% above the base line.

Another integrated approach is taken by Deshmukh and Allison (2017). They optimize a WPP system including WPP layout as well as WPP control facilitated by active wake control over the entire lifetime of the WPP. The optimal WPP system design is pursued using a quasi-steady empirical wake model (i.e. a deterministic wake, which expands downstream). The quasi-steady wake model is linearly superimposed on the undisturbed ambient flow fields including both mean wind shear and turbulence (presumably using only one turbulence seed and thereby one realization of the ambient stochastic turbulence field) to obtain a description of the WPP flow field. Surprisingly, a relationship between the atmospheric boundary layer (ABL) turbulence field and the introduced wake expansion factor is not established, although there is evidence that wake meandering, which depends on the site ambient turbulence field, is dictating the static downstream wake 'envelope' (Machefaux et al., 2015). Deshmukh and Allison (2017) take a model predictive control (MPC) approach that is specifically implemented using reduced-order state-space models of the individual WTs, which account for the tower for-aft bending dynamics, the rotor rotational speed dynamics and the blade pitch dynamics. The active wake control includes both WT derating and wake deflection by WT yawing. The objective function is the WPP annual energy production (AEP), and two case studies indicate a significant improvement of the integrated system design compared to layout design only. No attempt was done to compare the full integrated system design approach with a sequential approach, in which first the layout was optimized and then, subsequently, the WPP control. Such a comparison would have contributed to quantification of the coupling terms, elaborated by Fathy et al. (2001), and, in case of weak coupling, facilitated a reduction the computational efforts needed to perform the system design optimization. The paper is ended with a comparison of the relative AEP effect of the derating and the yaw-wake-deflection strategy, respectively. It is concluded that wake deflection is of marginal importance compared to WT derating.

Based on a two-WT case, Andersen (2019) analyzed active wake control using a high-fidelity CFD LES solver, fully coupled with a modal-based aero-elastic tool including a full dynamic WT controller. Comparing a 35° yaw case with the corresponding derating case, this study concludes that, for a given reduction of the upstream WT thrust, the yaw wake deflection strategy reduces the power production of the upstream WT more than the derating strategy. It is further concluded that the overall benefit of active wake deflection as well as WT derating is largely uncertain for a two-WT system.

Gebraad et. al (2015) also analyzed a two-WT case by means of high-fidelity CFD simulations of the wind farm flow field coupled with simulations of the WT dynamics. They used the National Renewable Energy Laboratory's simulator for wind farm applications (SOWFA) to investigate two different derating strategies: 1) changing the collective blade pitch setting; and 2) changing the tip speed ratio. They found that the power increase of the wake affected WT was balanced by the decrease of the derated WT. This lead to the conclusion that yaw wake deflection is more efficient than derating when quantified in terms of power production. Unfortunately, both of the investigated derating strategies are sub-optimal as shown by Vitulli et al. (2019). The optimal derating strategy is the particular combination of collective pitch- and tip speed setting resulting in the lowest rotor thrust for a given WT power production. Using this optimal derating strategy, Vitulli et al. (2019) obtained a considerable power gain.

Interestingly regarding both numerical two-WT studies by Andersen (2019) and Gebraad et. al (2015) is, in particular, a full-scale study by van der Hoek et al. (2019), in which derating of the most upstream WT was investigated for a single row of five WTs. Their CFD simulations predicted a power increase of 5.6% for the row-aligned wind direction in the below-rated wind speed regime. This result was verified by a year-long field test campaign, where WT derating was turned on and off every half week. The derating was, for practical reasons, implemented as two pitch offsets (one for full wake and one for partial wake conditions). The results of the field test predicted an increase of 3.3 % in AEP, i.e. close to the CFD simulations when taking the suboptimal pitch regulation as well as model and measurement uncertainties into account. Given that derating, based on pitch regulation only, is sub-optimal, there is a potential for even larger gain by using the optimal combination of pitch- and tip speed settings.

Large uncertainties, associated with both active wake control strategies (i.e. derating and yaw-based wake deflection) and, not least, among various simulation approaches as well as measurements, were also reported in (Kheirabadi and Nagamune, 2019). Important conclusions from this study is further, that 1) full-scale tests provide the most conservative (i.e. less optimistic) evaluations of the potential of active wake control; and consistently 2) *"added layers of realism in terms of simulated wind conditions tend to deteriorate the performance of wind farm controllers"*.

Guided by 1) and 2), the present contribution to WPP system design optimization (i.e. WPP layout- and control optimization) will seek to describe the complex inter-turbine aerodynamic interactions within a WPP as realistic as possible considering the computational resources needed for WPP optimization. This is done using an extremely fast full-blown CFD solver.

We will limit the scope to AEP[1] system optimization, i.e. WT loading is excluded. We assume that WT characteristics for aggregated AEP estimates are sufficiently described in terms of their power- and thrust coefficients, which implicitly include the relevant structural dynamics of a particular WT as e.g. crucial blade bending- and torsion dynamics for big modern WTs with flexible blades. Encouraged by the results obtained by Deshmukh and Allison (2017), Andersen (2019) and van der Hoek et al. (2019) we will limit WPP active wake control to WT derating and leave inclusion of yaw dictated wake deflection for a future study.

The research challenges dealt with in the present paper can be summarized as:

---

[1] Restricting the objective function to power production is a major simplification compared to approach taken in (Réthoré et al, 2013; Larsen and Réthoré, 2013; Mahulja et al., 2018) because: 1) aeroelastic modelling of the WPP WTs are circumvented; 2) a *stationary* description of the wake affected WPP flow field suffices; and 3) no cost models are needed.

1) Investigate WPP system optimization based on full-blown CFD simulations of the complex WPP flow field with its complicated WT wakes interactions;

2) Analyze and indicate the importance of the system coupling terms mentioned in (Fathy, 2001) – or more specifically their gradients with respect to the WT positions; and

3) Evaluate the AEP improvement potential accompanying the integrated system approach with a focus on individual WT derating based on analysis of an existing offshore WPP.

Sect. 2 describes the simulation platform including all relevant models, while Sect. 3 presents a simple and illustrative application example as a sanity check. The Lillgrund case study is described in Sect. 4. First the layout/control coupling is analyzed by a one-row WPP example. Based on the results of this study, a system optimization of the Lillgrund WPP is subsequently performed. The paper is concluded in Sect. 5, where also future work is identified.

## 2. The platform

Overall, the integrated layout and WPP control optimization platform is based on a fusion of Topfarm2 (2019), the DTU wake framework, PyWake (2019), and a dedicated aerodynamic rotor model. Topfarm2, which is the DTU open source WPP optimization framework, utilizes the open-source framework for multidisciplinary design, analysis and optimization, OpenMDAO (Gray et al. 2019), to find the optimal set of design variables, i.e. WT positions and control settings in a sequential or nested workflow. PyWake is the DTU open source AEP calculator including a collection of stationary wake models. PyWake is used to establish the AEP objective function needed in Topfarm2, which in this study is based on the linearized CFD RANS wake model, Fuga (Ott et al. 2011).

A simplified version of the present platform, excluding WPP layout optimization and thus only including WPP control optimization, is described by Vitulli et al. (2019). In its most general formulation, this open-loop WPP control optimization platform deals with two design parameters per WT – the tip speed ratio, $\lambda$, and the collective pitch angle, $\alpha$, both conditioned on the wind direction and wind speed. However, using a show case Vitulli et al. (2019) justifies that the design space, without loss of generality, consistently can be collapsed to only one parameter for each WT. This parameter reflects the desired derating and maps to a unique combination of collective pitch and tip speed ratio, $(\alpha_{Ct}, \lambda_{Ct})$, which results in the smallest possible thrust coefficient, $C_t$, conditioned on the requested power coefficient, $C_p$. For the sake of efficiency, we will take advantage of this finding in designing the present platform, thus resulting in three design parameters for each WT – two layout coordinates and the unique set $(\alpha_{Ct}, \lambda_{Ct})$ resulting from the unique functional relationship $\alpha_{Ct}(\lambda_{Ct})$.

In summary, the present integrated system optimization platform consist of four main components:

1) A CFD solver modelling the steady flow field within a WPP. The ambient mean wind shear and turbulence characteristics are specified in terms of a terrain roughness height conditioned on wind direction, which implicitly dictates the ambient turbulence conditions via the turbulence closure of the CFD model;

2) An aerodynamic part, which models the WT power- and thrust characteristics based on a detailed aeroelastic model of the WT. This model incorporates a description of both structural and aerodynamic properties of the WT with predefined settings for rotational speed and the collective pitch angle conditioned on the rotor inflow conditions. However, only steady WT deflections is accounted for in defining the rotor aerodynamic characteristics for the present purpose. This model is in turn used to establish an accurate and fast surrogate model to facilitate an efficient optimization process;

3) A WPP AEP performance metric defining the optimization objective, including possible constraints, as based on a priori available information on the mean wind direction probability density function (pdf) and the mean wind speed pdf conditioned on the wind direction of the site; and

4) An optimization platform that computes the optimal system performance in terms of WPP AEP metric while satisfying the site area and minimum wind turbine separation constraints.

In the following each of these four key elements are described in some detail.

## 2.1. The CFD solver

Typically, an optimization of the control settings for a WPP requires 200 - 1000 power evaluations for each mean wind speed and direction. To calculate a proper AEP metric, we use 23 speeds and 360 directions; i.e. 1.6 – 8.2 million flow field computations is needed to optimize the control settings for a given WPP layout. Obviously, this puts excessively high demands on the computational speed of the flow solver.

The linear CFD RANS solver, Fuga (Ott et al., 2011), is extremely fast, has previously compared well with full-scale
measurements (Peña et al. 2018, van der Laan et al. 2019), and is thus considered ideal for this task. The governing Navier-Stokes equations, neglecting the Coriolis forcing, are consistently linearized using a formal perturbation expansion and subsequently retaining only the first order perturbation terms. Thus, mass conservation is identically satisfied, momentum conservation is satisfied to first order, and the resulting WPP fields are divergence free, as they should be for an assumed incompressible flow. The resulting equations are in turn conveniently formulated and solved in a mixed-spectral domain for
efficiency reasons. The velocity perturbation around a single WT in the physical domain is derived from Fourier components of the mixed-spectral solution using a fast inverse Fourier integral transform and stored in a system consisting of both general and WT-specific look-up tables, which facilitates the extreme computational speed of the solver. Because of the linearity of the model, wakes from multiple upstream WTs can consistently be superimposed to construct the flow field further downstream. From an efficiency perspective, this is a big advantage.

The WTs are modelled as actuator discs, which in general can be vertically inhomogeneous, but is often assumed uniform in wake studies. The actuator discs embedded in the flow field represent the rotor drag forces, which in turn is responsible for creation of rotor downstream wakes. The specifications of the individual actuator discs are based on detailed aerodynamic models of the WPP rotors as accounted for in Sect. 2.2. The WPP wind field, impinging at an arbitrary WT in the WPP, depends on the ambient wind field and wakes from relevant upstream WTs linearly superimposed.

The inflow conditions, i.e. mean wind speed and direction, are assumed horizontally homogeneous over the spatial extend of the WPP. More specifically, neutral atmospheric boundary conditions are assumed, meaning that a logarithmic mean wind shear profile applies. The characteristics of the shear profile is thus in turn defined by a terrain-roughness length and the friction velocity $u_*$. For neutral atmospheric conditions, Monin–Obukhov scaling dictates the standard deviation of the velocity fluctuations to be invariant through the atmospheric boundary layer and proportional to the friction velocity. The turbulence
inflow is thus expressed in terms of the same input parameters as the mean wind shear field. For the Lillgrund site, a roughness length of $z_0 = 0.03$ m is used, which results in an inflow turbulence intensity of approximately 12%. This is in the high end for an offshore location and relates to the proximity of the Lillgrund site to urban areas. No attempt was done to link the roughness length to inflow wind speed, because site measurements have shown only marginal variations of the turbulence intensity with wind speeds below the rated wind speed.

For each wind direction, the local wind speed, i.e. ambient wind speed minus the sum of deficits from upstream turbines, the power production and the thrust coefficient as well as the wake deficits at downstream WT positions are evaluated starting with the most upstream WT position and continuing in the downstream order.

## 2.2. The aerodynamic WT model

As mentioned, we consider detailed aerodynamic rotor performance expressed in terms of power- and thrust coefficients as
fully satisfactory for WT AEP simulations.

Initially, the power- and thrust coefficients of the rotor is modelled using HAWCStab2 – a linearized aero-servo-elastic code designed for stability analysis and steady-state simulation of WTs (Hansen et al., 2017). HAWCStab2 relies on an extended formulation of the traditional blade element momentum (BEM) approach (Madsen et al., 2007), and consequently detailed geometric- and aerodynamic input is required, e.g. the blade planform and twist distribution as well as blade aerodynamic properties in terms of aerodynamic coefficients over the blade length. In the present application, HAWCStab2 uses a fully flexible WT model formulation to account for the equilibrium-static wind-speed-dependent deflections of the WT main components and thus the potential effects on the WT thrust- and power performance.

For traditional layout optimization without WPP control, the WPP production is implicitly based on WTs running at maximum $C_p$. For the present application, which aims at system optimal design, the aerodynamic modelling includes a WT derating feature, which links to a unique set of tip speed ratio and collective pitch angle, $(\alpha_{Ct}, \lambda_{Ct})$. Consequently, the aerodynamic WT model must facilitate computation of $C_t$ and $C_p$ conditioned on these design variables. Assuming zero yaw error, the tip speed ratio, $\lambda$, is defined as

$$\lambda \equiv \frac{R\Omega}{U} \tag{1}$$

where $R$ is the rotor radius, $\Omega$ denotes the rotor speed, and $U$ is the hub height mean wind speed.

The conditional dimensionless rotor thrust and power coefficients are defined as respectively

$$C_t(U|\alpha, \lambda) \equiv \frac{T_{WT}(U|\alpha, \lambda)}{\frac{1}{2}\rho A U^2} \tag{2}$$

and

$$C_p(U|\alpha, \lambda) \equiv \frac{P_{WT}(U|\alpha, \lambda)}{\frac{1}{2}\rho A U^3} \tag{3}$$

where $T_{WT}$ is the rotor thrust force, $P_{WT}$ is WT power production, $\rho$ is the air density, and $A$ is the rotor area, which depends on both the rotor tilt ($\theta_t$) and the blade coning ($\theta_c$) angles as

$$A = \pi(R \cos\theta_c \cos\theta_t)^2 \tag{4}$$

In this context, $P_{WT}$ and $T_{WT}$ is obtained from HAWCStab2 simulations of the Siemens SWT-2.3-93 WT, which is operating at the Lillgrund WPP; see Sect. 4. The steady-state power and thrust have been simulated for a range of collective pitch and rotor-speed settings in a uniform flow field of 8 m s$^{-1}$. In principle such steady-state parameter sweep simulations must be performed for all relevant mean wind speeds to account for the steady-state blade deflections. However, assuming that these deflections have only a minor effect on the steady-state power and thrust performance for the WT in question, then one mean wind speed suffices. This is justified under the assumption that the thrust scales with $U^2$ and the thrust coefficient is normalized with $U^2$, whereas the power scales with $U^3$ and the power coefficient is normalized with $U^3$.

Note from eq. (1) that for a fixed wind speed, a variation in rotor speed corresponds to a variation of $\lambda$. Thus, from the above described simulation outputs, the power and thrust coefficients are easily calculated as a function of the tip speed ratio and the collective pitch via equations (1) - (4); see Figure 1.

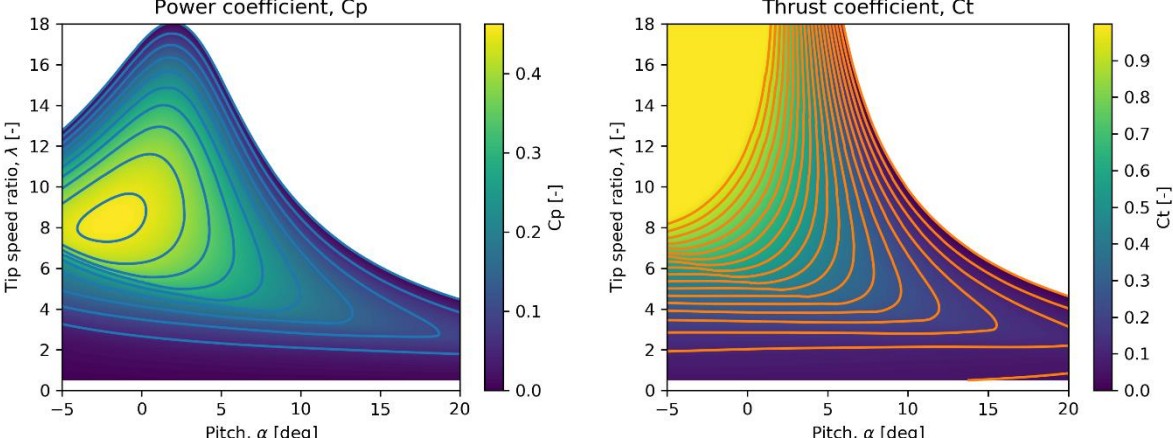

**Figure 1. Power- and thrust coefficients as a function of tip speed ratio and collective pitch angle, based on HAWCStab2 simulations of a Siemens SWT-2.3-93 WT.**

The results shown in Figure 1 can be used for the entire range of mean wind speeds requested for the system optimization, cf.

eq. (1). This is convenient from a computational point of view, and thus consolidates the tip speed ratio as design variable as an appropriate choice.

Another important computational simplification is, as previously mentioned, the reduction from two control design variables per WT to one control design variable per WT. This reduction is based on the previously mentioned findings by Vitulli et al. (2019) showing that optimal derating is obtained by selecting the unique set of design variables, $(\alpha_{Ct}, \lambda_{Ct})$ which, for a given

derating (i.e. power production reduction), corresponds to the smallest possible thrust. This condition, which is also intuitively clear, provides a unique relationship between $\alpha_{Ct}$ and $\lambda_{Ct}$ and justifies the reduction in design space to one control variable per WT, conditioned on ambient mean wind direction and mean wind speed.

As a consequence of the control design space collapse, a specific derating factor corresponds to a deterministic path through the original $(\alpha, \lambda)$ design space, where the points on this path correspond to certain mean wind speeds. Note, that these paths

are constrained by the minimum and maximum rotor speed limits as well as the maximum power limit, see Figure 2.

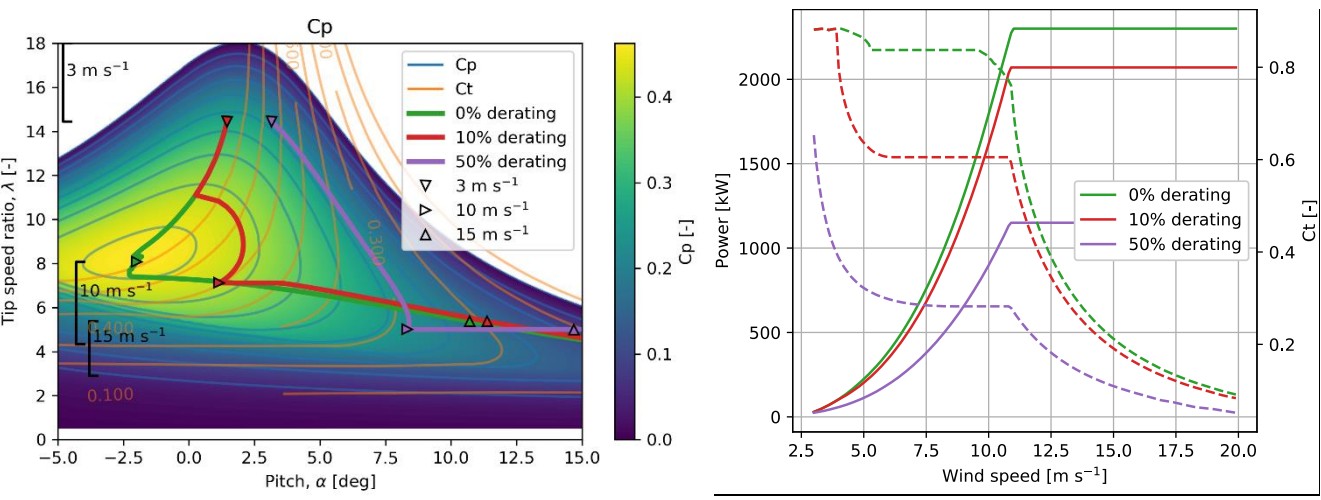

**Figure 2. Left: C$_p$, (background colour and blue contours) and C$_t$ (orange contours) plotted as a function of tip speed ratio, λ, and collective pitch setting, α. The green, red and purple lines expose the ($\alpha_{Ct}$, $\lambda_{Ct}$)-relation for 0%, 10% and 50% derating, respectively.**

**These relations are plotted for a range of wind speeds (3, 10 and 15 m s$^{-1}$ is marked) satisfying the rotor speed limits (indicated in the left-hand side of the figure for 3, 10 and 15 m s$^{-1}$) as well as the maximum power limit. Right: The corresponding power (solid) and C$_t$ (dashed) curves plotted as a function of wind speed. These figures are based on HAWCStab2 simulations of a Siemens SWT-2.3-93 WT model.**

The last step needed to prepare for an efficient optimization procedure is to transform the above described aerodynamic rotor computations into a surrogate model, which maps mean hub wind speed and the requested derating factor into a power production- and a thrust coefficient conditioned on the operational settings; i.e. $C_p (U|\alpha,\lambda)$ and $C_t (U|\alpha,\lambda)$. The surrogate thereby establishes the link between the derating settings, to be specified by the control optimizer, and the characteristics of the uniformly loaded actuator discs needed by the flow solver. Note, that controller-specific constraints such as tower-exclusion zone and smooth transition between regions as well as controller implementation issues are not taken into account. At present, it was not found essential to model the actuator discs as vertically in-homogeneous, although possible within the framework.

## 2.3. The AEP performance metric and constraints

The objective function defined for the present optimization platform is the AEP of the WPP. Financial costs of internal WPP power grid etc. are not considered, which in turn means that the positions of the individual WPP WTs are only constrained by the minimum allowable distance to the nearest neighboring WT and the line of demarcation defining the permissible WPP area. Considering 2 rotor diameters ($D$s) to be the minimum realistically WT interspacing distance, this minimum spacing constraint has been selected for all show cases presented in this paper. The permissible WPP area for the Lillgrund case is the stylized convex shape of the Lillgrund reef. Finally, we have incorporated additional two constraints associated with the operational conditions of the Lillgrund Siemens SWT-2.3-93 WT used in all cases: $\Omega \in$ [9 rpm; 16 rpm] and $\alpha \in$ [-2°; 90°].

In each iteration of the optimization procedure, the objective function – in this case the AEP performance metric – must be computed. Computational efficiency is in particular needed for the present CFD-based approach, and maximum efficiency is assured through implementation of the 'shortcuts' described in Sect. 2.2.

For a given layout (i.e. associated with a given iterative step in the layout optimization process), the WPP AEP, $P_{AEP}$, is estimated from

$$P_{AEP} = T \sum_{i=1}^{N} \int_{0°}^{360°} \int_{U_{in}}^{U_{out}} P_i(U|\theta) f_U(U|\theta) f_\theta(\theta) dU d\theta \tag{5}$$

in which $U$ denotes the undisturbed ambient hub-height mean wind speed and $P_i (U|\Theta)$ is the production (in watt) of the $i$th WT at ambient hub-height mean wind speed, $U$, and associated operating conditions dictated by the internal WPP flow field. $f_U (U|\Theta)$ is the ambient hub-height mean wind speed pdf, conditioned on the ambient mean wind direction (i.e. often a two parameter Weibull distribution), and $f_\Theta (\Theta)$ is the ambient mean wind direction pdf. Assuming SI units, $T$ is the number of seconds corresponding to one year, and $N$ is the pre-defined number of WTs within the WPP considered.

In practice, eq. (5) is discretized to facilitate evaluation of the involved integrals. In the succeeding study cases, a directional discretization of 1° was used combined with an ambient mean wind speed discretization of 1 m s⁻¹.

## 2.4. Optimization setup

Overall, there is three common ways to design the WPP system optimization. The most elaborate of these is to design the full integrated approach by involving all design variables simultaneously - the *one-step* approach. The layout optimization related design variables amount to two (i.e. the WT position in a Cartesian coordinate system) per WT. The WPP control optimization, conditioned on ambient mean wind direction and mean wind speed, requires, utilizing the design space collapse described in Sect. 2.2, one design variable per WT. However, because the AEP computation requires all wind directions and all wind speeds to be accounted for, the control-related design variables amounts to $N_d N_s$ per WT. Here $N_d$ is the number of ambient inflow directions, and $N_s$ is the number of ambient mean wind speeds considered in the discrete version of eq. (5). Thus, in total the number of design variables amounts to $N(2+N_d N_s)$. This is clearly infeasible within the present framework – even when utilising a high-performance-computing cluster.

An alternative and more efficient strategy for a full integrated system optimization is a *two-step nested* approach, in which, for each optimization step, first the layout is advanced and then, based on this iteration of the layout, the associated optimal control schedule, conditioned on ambient mean wind speed and direction, is determined. Merging the sequentially determined WPP layout and associated optimal control schedule, the AEP estimate, associated with the actual iterative step, can be evaluated.

The associated workflow is illustrated in Figure 3.

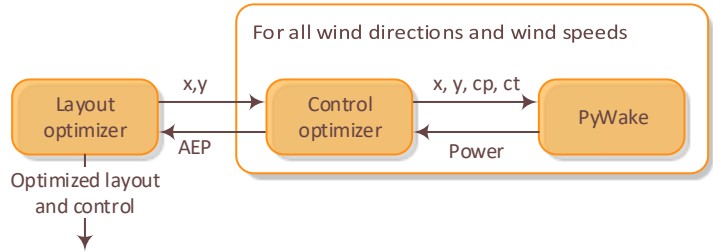

**Figure 3. Nested optimization workflow. The control settings are optimized in every layout iteration.**

Both the one-step optimization strategy and the two-step nested optimization strategy are fully integrated strategies, which eventually will lead to the same result.

If the optimal system design is separable, in the sense that only a weak coupling exists between the layout- and the WPP control optimization, the problem can be significantly simplified. This will be quantified in Sect. 3.3 and Sect. 4.1 using two demonstration cases. The significant reduction in computational complexity is obtained taking a *two-step sequential* approach by approximating a weak system coupling with no system coupling. The sequential workflow, in which the conventional 'greedy' individual WT control settings are used for the WPP layout optimization, is succeeded by an optimization of the WPP

control scheduling conditioned on both ambient mean wind speed and direction. Thereby, the 'greedy' WT control settings are replaced by optimized 'collaborative' WT settings to the benefit of the WPP AEP. The workflow associated with this sequential strategy is shown in Figure 4.

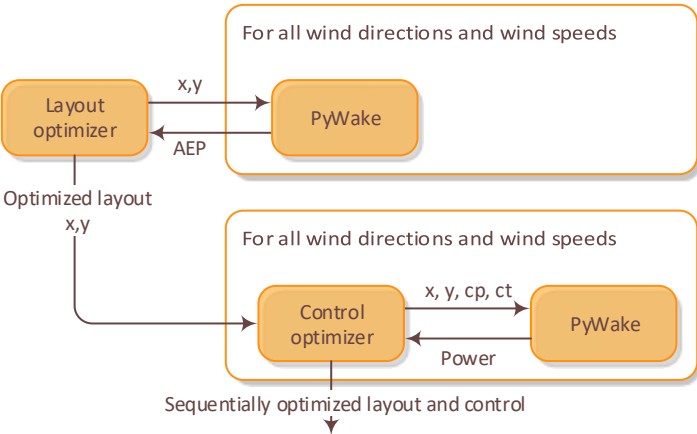

**Figure 4. Sequential optimization workflow. The control settings are optimized one time only, after the optimal layout is found.**

The merger of these two optimization steps makes up the optimized system design and is in essence a sequential application of the Topfarm2 (2019) layout platform and the open-loop WPP control scheduling platform described by Vitulli et al. (2019). The layout is optimized using a combination of random search and gradient-based (SLSQP) optimization. The random search algorithm does not get stuck at local optima and is consequently suitable to find a good global solution, while the gradient-

based optimizer, applied subsequently, is used to trim the random-search solution to the nearest optima. In this setup, the gradients are approximated by a finite difference approach. For a complex non-convex optimization problem, a global optimum

cannot theoretically be ensured, but running numerous random sequences converging to almost identical results gives confidence to the result being close to the global optimum.

The WPP control scheduling optimization problem has in general only a few local optima and can therefore easily be solved by the gradient-based optimizer using gradients computed via finite difference. This control optimization is, however, rather time consuming, as the WT control settings must be optimized for all 360 x 23 combinations of wind directions and wind speeds; see Table 1. These combinations are, fortunately, independent, and the workflow therefore suitable for parallel computation. For the current study, a parallel workflow utilizing 360 CPUs (i.e. corresponding to a 1° mean wind direction resolution) has been set up, where each CPU optimizes all WT control settings for one wind direction. Table 1 gives an idea of the computational resources needed for the case studies performed in Sect. 3 and Sect.4.

|  | Row of 8 WT (1D layout, 1 WD, 23 WS) | Lillgrund, 48 WT, (2D layout, 360 WD, 23 WS) |
|---|---|---|
| PyWake, AEP calculation | 0.002s | 0.52s |
| Control optimization | 3.5s | 15h (1 CPU) 4 min (360 CPU) |
| Layout optimization | 3.4s | 2.8h (random search; 1 CPU) + 1.2h (gradient based; 1 CPU) |

**Table 1. Overview of time consumption of the AEP calculation, the control optimization and the layout optimization.**

## 3.   Sanity check

To check the overall behaviour of the optimizers, a sanity check on a simple illustrative example, consisting of a row with three Siemens SWT-2.3-93 WTs, has been performed. This case is selected, because it can be solved via 'brute force', and because the results are easily visualized.

### 3.1. Control optimization

The sanity check of the control optimization is performed on a simple example consisting of three WTs on a row, separated by $4D$, and with a uniform inflow of 10 m s$^{-1}$ aligned with the row; see Figure 5.

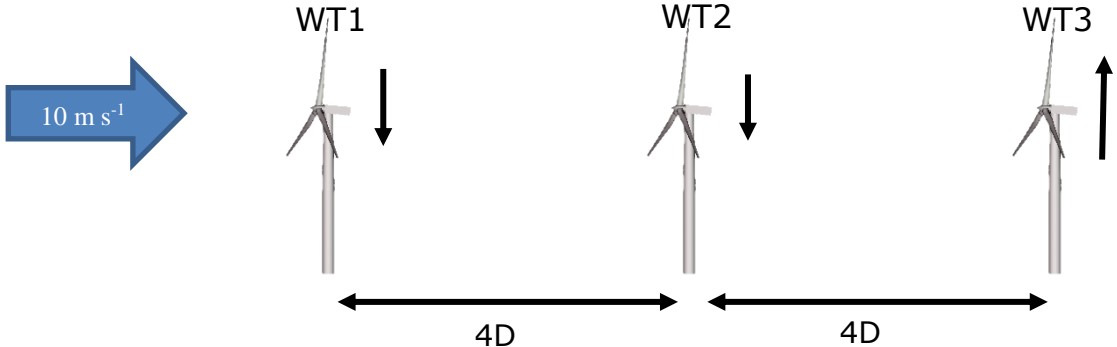

**Figure 5. Three-WT row used for sanity check of the control optimization. Reduced WT production, caused by derating, is indicated by an arrow pointing down - increased WT production, caused by optimized WPP control, is indicated by an arrow pointing upwards.**

Figure 6 shows the power produced by the three WTs as a function of the derating of the two upstream WTs. In the left plot, it is seen that the power for the most upstream WT, WT1, only depends on its own derating setting. The power of WT2, on the other hand, depends on the derating of both itself and of WT1. Finally, it is seen that WT3, obviously, produces the most, if both WT1 and WT2 are derated 100%.

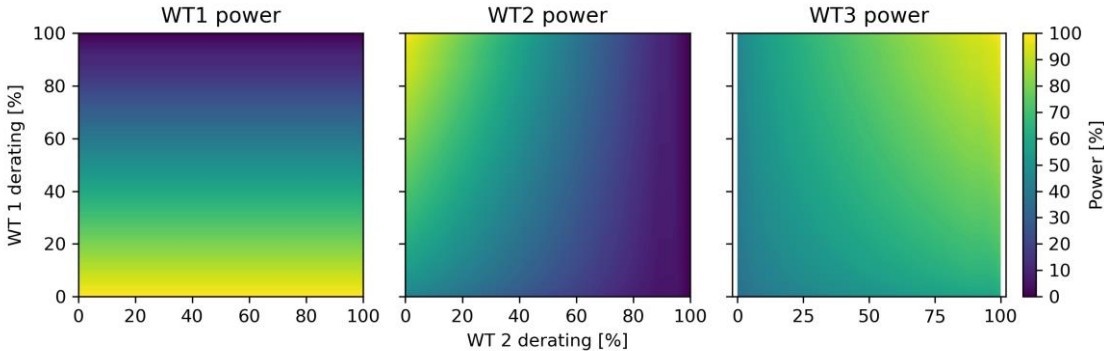

Figure 6. Power produced by the three WTs in 10 m s⁻¹ as a function of WT1- and WT2 derating.

The total power produced by the three WTs is seen in Figure 7, and it appears that the total power can be increased by 4.01% if WT1 is derated by 7%, and WT2 is derated by 5%.

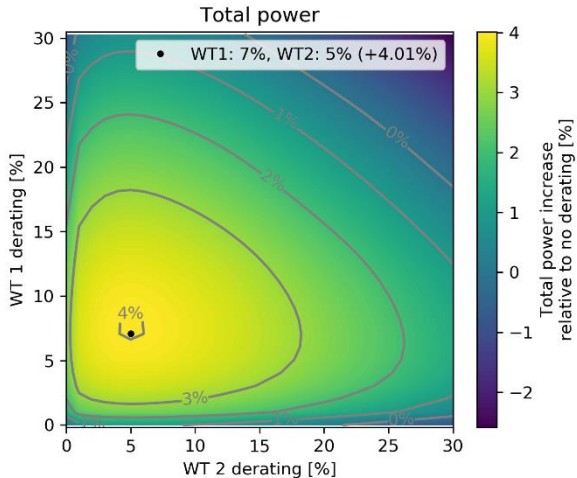

**Figure 7. Total power of the three-WT row as a function of the derating of WT1 and WT2. The power can be increased by 4.01% when WT1 is derated by 7% and WT2 is derated by 5%.**

### 3.2. Layout optimization

A sanity check of the layout optimizer is also performed on the three-WT row. In this case, the position of WT2 is allowed to vary between $2D$ and $6D$ behind WT1. Figure 8 shows the individual relative power production of the three WTs as well as the total power production as a function of the position of WT2 in a uniform flow of 10 m s⁻¹ aligned with the row. As expected, WT1 is unaffected by the position of WT2, while the power production of WT2 increases with the distance to WT1, and vice versa for the power of WT3. Finally, the total power production is seen to increase slightly, when WT2 is moved downstream.

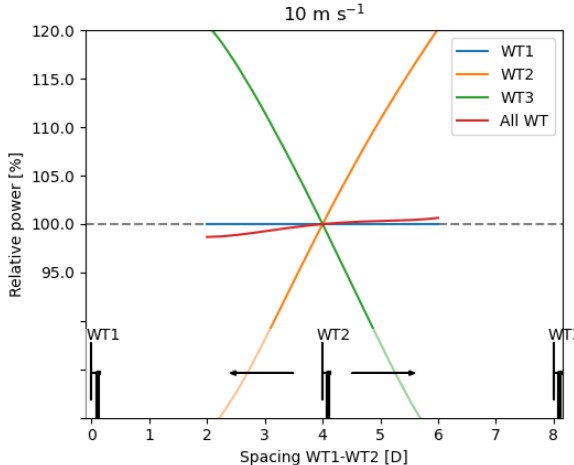

**Figure 8. Relative power produced by the three individual WTs as well as the total relative power plotted as a function of the position of WT2.**

For other wind speeds, however, the picture is quite different, as seen in Figure 9. The optimal position thereby depends on the wind speed distribution, which links to the dependence of $C_p$ and $C_t$ on wind speed with the hub-height mean wind speed. Plotting the relative AEP computed using the Weibull distribution associated with westerly winds at the Lillgrund wind farm (c.f. the wind rose shown in Figure 12) reveals, that the optimal spacing, under these conditions, is very close to 4*D*.

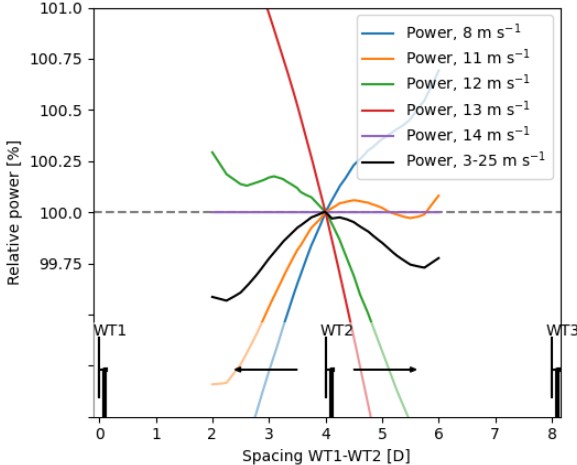

**Figure 9. Relative total power production of the three WTs plotted as a function of the position of WT2 for different wind speeds. The power for 3-25 m s$^{-1}$ is weighted by the Weibull distribution associated with wind from 270 °**

### 3.3. Combined layout and control optimization

The performance of the integrated layout- and control optimization is illustrated in Figure 10. The blue line indicates the relative AEP of the three WTs as a function of the position of WT2 in case all WTs are operated 'greedy' (i.e. no derating). This is the base case. The optimal position of WT2 is found to be 3.96*D* downstream of WT1. Applying layout dependent optimal derating of WT1 and WT2 (orange curve) sequentially increases the AEP of the initial layout by 2.221%. Finally, the AEP is seen to increase only infinitesimally (i.e. increasing from +2.221% to +2.226%), when applying integrated *two-step nested* system optimization. For the investigated simplistic case, this result indicates a very weak system coupling between WPP layout- and control optimization.

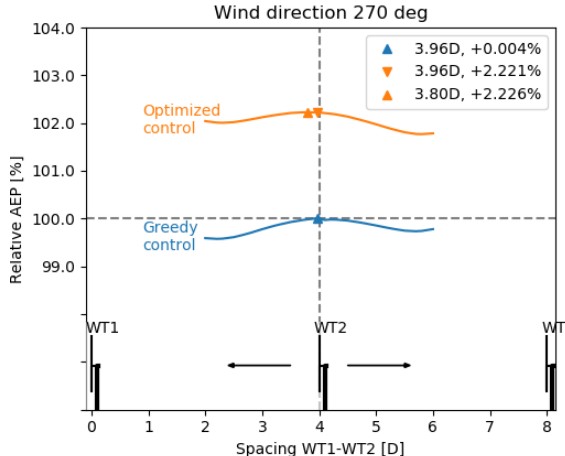

**Figure 10. Relative AEP plotted as a function of the position of WT2 for both greedy and optimized control.**

## 4. The Lillgrund showcase

The Lillgrund WPP is located in Øresund between Denmark and Sweden and consists of 48 Siemens SWT-2.3-93 WTs with a rotor diameter of 93 m. The WPP is known for its very small WT interspacings, down to 3.3$D$ and associated pronounced wake effects. This makes this WPP especially suited for studies of WPP performance. The WPP layout is shown in Figure 11.

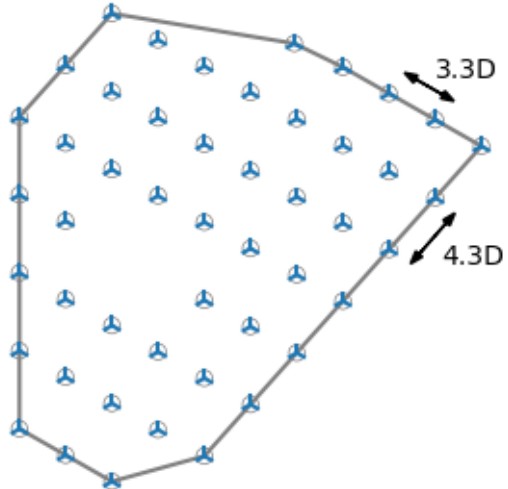

**Figure 11. WT positions in the offshore Lillgrund WPP.**

The Lillgrund wind climate is outlined in Appendix A in terms of ambient mean wind speed pdfs (i.e. two-parameter Weibull), conditioned on the ambient mean wind direction as well as an ambient mean wind direction pdf. For the sake of illustration, the applied wind climate information is condensed in the wind rose shown in Figure 12, which reveals predominant winds from west and south.

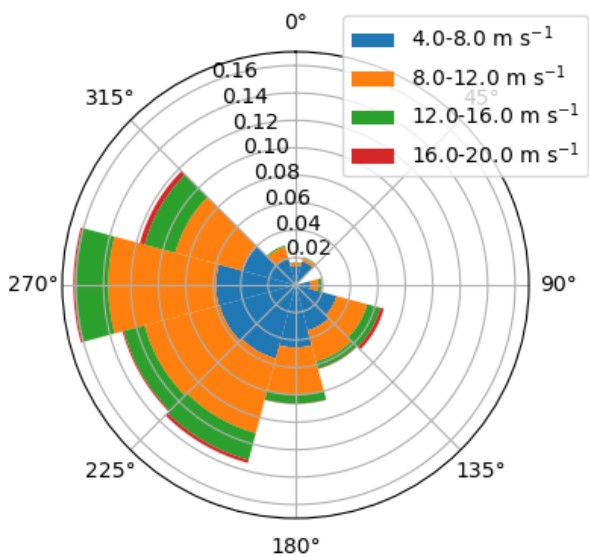

**Figure 12. Wind rose characterizing the wind climate at the Lillgrund wind farm. Mean wind speed bins are shown in different colours, and their occurrence probabilities (conditioned on the respective inflow sectors) are proportional to their respective radial extend.**

First, we will focus on a subset of the Lillgrund WPP consisting of a row of eight WTs with along-row inflow conditions covering the entire relevant wind speed regime – i.e. the wind speed regime within which these WTs are in normal operation. Using this simplified case study, we will investigate the system coupling between WPP layout- and WPP control optimization. Based on the results from this study, we will next perform a system optimization of the Lillgrund WPP and thereby quantify its potential in terms of increased AEP compared to the base case, which is the present layout (cf. Figure 11) without

coordinated WPP control – i.e. only conventional 'greedy' control of the individual WTs.

## 4.1. Eight-WT row

This case study basically consist of one of the three Lillgrund WWP rows with eight WTs, meaning that the WT interspacing in the base case is 3.3$D$ (cf. Figure 11), and that the WTs are Siemens SWT-2.3-93. The wind climate is fictitious, as only an along row inflow direction is considered, which assures largest possible mutual WT wake interactions. Within this framework

we have, without loss of generality, assumed Weibull distributed mean wind speeds corresponding to the 270° site condition (however, truncated to the relevant wind speed regime [3 m s$^{-1}$; 25 m s$^{-1}$]) although the 'true' inflow direction associated with this row is 300°.

With the purpose of investigating the strength of the system coupling, we have optimized: 1) the WPP layout; 2) the WPP control; 3) the integrated WPP layout and WPP control based on the *two-step sequential* approach (cf. Sect. 2.4); and 4) the

20 integrated WPP layout and WPP control based on the *two-step nested* approach (cf. Sect. 2.4). Based on a pre-investigation of optimizers, where the 'random search' approach was compared to the SLSQP gradient-based optimization algorithm, the latter was found clearly superior and consequently used in this study.

The results of the investigation are, together with the base case (0), summarized in Table 2.

| Case | Layout | Control | AEP [Gwh] | CPU time [s] | |
|---|---|---|---|---|---|
| 0 | Initial | Greedy | 40.85 | 0.01 | 0% 0% 0% 0% 0% 0% 0% 0% |

| | | | | | |
|---|---|---|---|---|---|
| 1 | Initial | Optimized | 44.10 (+8.0%) | 4.20 | 15% 19% 18% 16% 13% 10% 4% 0% |
| 2 | Optimized | Greedy | 41.44 (+1.4%) | 2.84 | 0% 0%  0%  0%  0%  0% 0% 0% |
| 3 | Optimized | Optimized(sequential) | 44.558 (+9.1%) | 6.92 | 18%16%  14%  12%  11%  11%4% 0% |
| 4 | Optimized | Optimized(nested) | 44.560 (+9.1%) | 3731 | 15%16%  14%  13%  11%  11%4% 0% |

**Table 2. AEP results of various optimization approached applied on the eight-WT row. The CPU times refer to the computation time on a standard laptop PC. The figures in the rightmost column show the position of the eight turbines. The derating settings of the WTs are indicated by the colour of the turbine symbol and quantified in percentage by the number above the WT symbols.**

The base case, case(0), represents the existing layout with conventional greedy control of the individual WTs. Case(1) represents the base case layout with the WPP control optimized. The associated increase in AEP, relative to the base case, is significant and amounts 8.0%. The close spacing in the Lillgrund WPP case ($3.3D$) is comparable with the WT inter spacing ($2.3D$ - $3.1D$) in the Goole Fields WPP investigated in (van der Hoek et al., 2019), where an increase of 5.6% for a row of five WTs was predicted, and an increase of 3.3% was realized in the accompanying full-scale study. As noted in the discussion of the results in this paper, the $k$-$\varepsilon$ turbulence closure of the CFD model, which was used for tuning of the derating settings, supposedly makes the CFD predictions underestimate wake effects for closely spaced WTs, whereby the used pitch settings are likely to be sub-optimal. Furthermore, only the first WT in the investigated row was derated, their two-step pitch-offset derating strategy was suboptimal, and finally the derating potential increases with the number of WTs. This, paired with the fact that full-scale studies always will suffer from imperfect inflow (wind direction variability within the 10-minute recording sequences etc.) and WT operational conditions (moderate yaw errors etc.) make us believe that the case(1) result is fairly consistent with the results presented in (van der Hoek et al., 2019). In case(2) the WT applies greedy control, and the WT row layout is optimized. The increase in AEP, relative to the base case, amounts to 1.4%, which is considerably less than achieved in case(1). Compared to the three-WT case in Figure 10, the AEP increase achieved by layout optimization in this case is much more pronounced because the number of design variables has increased from one to six. It is seen that the distance between the two most upstream and the tree most downstream WTs are smaller than in the base case. This allows larger spacing and thereby production of the middle turbines, which, in this case, results in an increase of the AEP of the whole row. Obviously, this strategy is not possible with only three WTs. Case(3) represents one of two system optimization approaches. Here we assume that the system optimization is separable and consequently can be performed by first optimizing the layout and subsequently the WPP control. The combined effect is an increase in AEP amounting to 9.1%, which is significant and exceeds what was obtained by only optimizing the WPP control (i.e. case(1)). In the second and last system optimization strategy, case(4), the integrated *two-step nested* approach is taken. Although being more complex and time consuming (around 540 times slower) than the case(3) strategy, the outcome is not significantly improved (cf. Table 2).

In conclusion, we have shown that the strength of the system coupling between WPP layout- and WPP control optimization is only marginal for the considered eight-WT case study characterized by 'heavy' mutual WT wake interactions.

## 4.2. Full Lillgrund wind farm

This case study comprises the entire Lillgrund WPP, and it eventually aims at quantifying the potential of an integrated system optimization of WPP layout and WPP control.

In analogy with Sect. 4.1, we will investigate a variety of WPP layout and WPP control optimization strategies. The control optimization schedule and the associated results appears from Table 3.

| Case | Layout | Control | AEP [Gwh] |
|------|--------|---------|-----------|
| 0 | Initial | Greedy | 345.2 |
| 1 | Initial | Optimized | 349.5 (+1.3%) |
| 2 | Optimized | Greedy | 354.9 (+2.8%) |
| 3 | Optimized | Optimized(sequential) | 358.7 (+4.0%) |

**Table 3. AEP results of various optimization approached applied on the full Lillgrund WPP.**

The investigated cases are analogous to the cases investigated for the eight-WT case in Sect. 4.1. As for case(1) we see a considerably drop in performance increase compared to the eight-WT situation, which is due to the persistently more severe mutual WT wake interactions in the fictitious eight-WT situation compared to the full Lillgrund WPP, where WT wake interactions for some inflow directions are limited (cf. Figure 14). With less wake interaction follows intuitively less potential for WPP control. Case(2) represent an isolated WPP layout optimization retaining the 'greedy' individual WT control performance. The associated increase in AEP performance amounts to 2.8% – or more than double that of the WPP control optimization, case(1). The last case, case(3), represents a system optimization approach. Based on the investigations performed in both Sect. 3.3 and Sect. 4.1, we assume that the system optimization is separable in the sense described in Sect. 4.1. The rationale justifying this assumption is, that the system coupling between WPP layout- and WPP control optimization was shown to be marginal in the eight-WT case, in which the overall WT wake interaction, over all inflow directions, are significantly more pronounced than for the full Lillgrund case. Taking the sequential approach, the combined Lillgrund WPP optimization results in an AEP improvement of 4.0%, which is significantly more than each of the individual layout and WPP control optimization approaches. Finally, it should be noted that, although possible, the *two-step nested* approach will require horrendous CPU resources and even on a cluster take in the order of a few months to conduct.

The layout resulting from the Lillgrund WPP system optimization is shown in Figure 13 together with the base line layout.

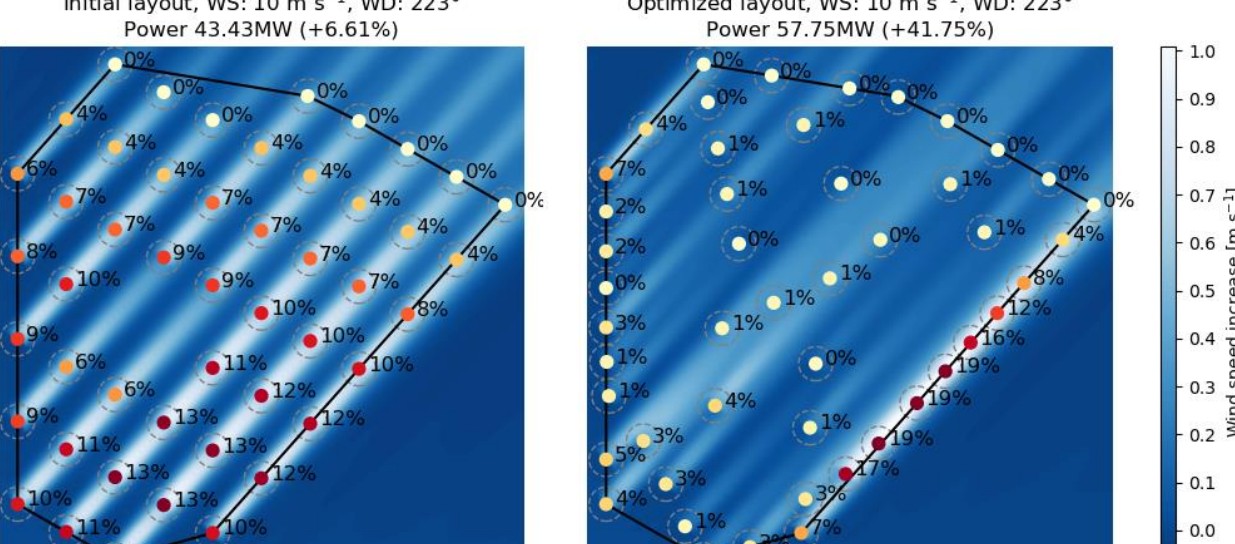

**Figure 13. The base line Lillgrund WPP layout (left) and optimized Lillgrund WPP layout (right). The two figures show the flow case associated with 10 m s⁻¹ inflow from direction 223°. The derating settings of the individual WTs are indicated by the colour of the WT symbols and quantified in percentage by the number above the WT symbols. The background colours illustrate the increase in wind speed from the individual 'greedy' to the 'collaborative' optimized control situation.**

From a pure production perspective, it makes sense to locate WTs densely at the boundary of the 'admitted area' for the WPP, because it intuitively will reduce the WT wake interactions. Notable is also that the individual WT deratings for the shown example, except for one row, is considerable less than for the base line case.

The results for all the investigated optimization strategies are summarized in Figure 14 and Figure 15. Figure 14 shows the increase in AEP conditioned on the inflow mean wind direction. As expected, the AEP gains vary with the wind direction with huge increases, up to 50%, for the optimized layout for the wind directions that is parallel to the rows of the original layout, i.e. 120°/300°, 42°/222° and 0°/180°. These increases, however, are almost balanced out by the decrease at other directions resulting in the average increase of the 2.8% and 4% increase that is reported in Table 3.

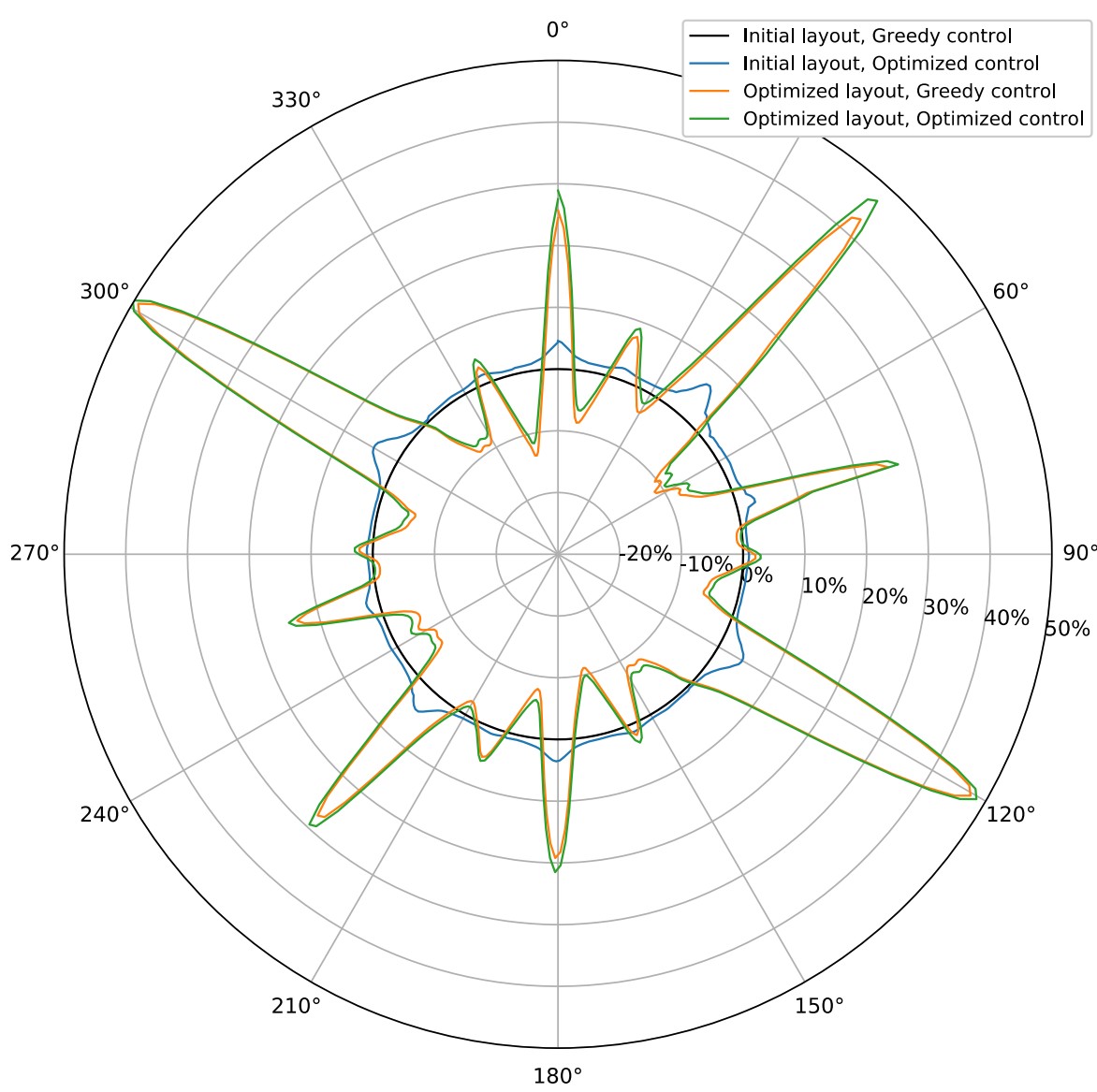

**Figure 14. Increase in AEP due to layout and/or control optimization plotted as a function of inflow wind direction.**

In Figure 15, the AEP gains are shown as a function of the mean inflow wind speed. The largest increases are seen below 10-11 m s⁻¹ where all WTs operate below rated power. At higher wind speed, the WPP production wake losses decreases, as more and more WT reaches rated power, thus eventually completely eliminating any WPP control potential.

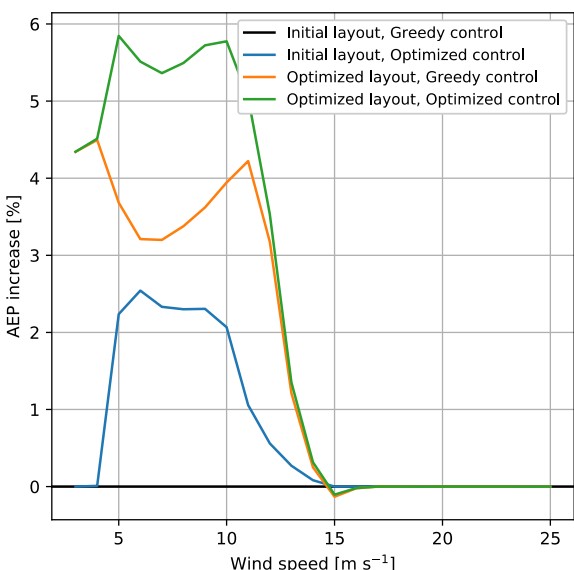

**Figure 15. Increase in AEP due to layout and/or control optimization plotted as a function of wind speed.**

## 5. Conclusions

This paper describes a platform for integrated WPP layout- and derating-based WPP-control optimization. The objective function for the optimization is the AEP of the WPP without considering financial costs of internal WPP grid etc. This means that the positions of the individual WPP WTs are only constrained by a minimum allowable distance to the nearest neighboring WT, in this case 2D, and the convex boundary around the initial WPP layout.

As WPP loading is excluded, stationary modeling of the complex WPP flow field suffices, which is a considerable simplification. Contrary to other known WPP optimization platforms, the present approach is based on a consistent and very fast CFD solver, whereby the inherent uncertainties associated with simple empirical algebraic wake models, including their often debatable wake summation 'receipt', is avoided. This strategy is consistent with a recent review of WPP optimization approaches (Kheirabadi and Nagamune, 2019), where one of the conclusions is that "*added layers of realism in terms of simulated wind conditions tend to deteriorate the performance of wind farm controllers*", thus stressing the importance of carefully and realistically simulated WPP flow fields.

The platform has initially successfully been subjected to a simplistic sanity check. Subsequently, the platform has been used to analyze the potential of an integrated WPP layout and WPP control optimization of the offshore WPP Lillgrund, which consists of 48 closely spaced WTs. First, an analysis of the system coupling between WPP layout optimization and WPP control optimization is performed as based on a subset of this WPP exposed to inflow conditions clearly exaggerating the overall complex inter-WT aerodynamic interactions within a traditional WPP, because all the WTs are in a state of maximum wake interaction. The study demonstrates an inferior system coupling only, thus justifying separation of the present optimal system design. Based on this learning, a full system optimization of the Lillgrund WPP is performed, resulting in a gain amounting to 4.0% in AEP relative to the base line case, which is the present Lillgrund layout without WPP control.

In a future perspective, the platform will be extended to also include active wake control in terms of WT yaw dictated wake deflection. This requires a generalization of the applied linearized CFD flow solver Fuga – a work that is in progress.

## 6. Data availability

Simulation data not available due to confidentiality of the Siemens WT model.

## 7. Author contribution

GCL has designed the numerical study. MMP implemented and run the optimizations. The paper was written and reviewed in cooperation.

## 8. Competing interests

The authors declare that they have no conflict of interest.

## 9. Acknowledgement

Financial support from the EU Horizon 2020 research and innovation program, under grant agreement no. 727680 (TotalControl), is acknowledged. Siemens Gamesa Renewable Energy is acknowledged for making the aerodynamic data of the Siemens SWT-2.3-93 WT available for the Lillgrund study.

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

**Appendix A**

| Wind sector (centered) [deg] | Frequency [%] | Weibull scale (A) | Weibull shape (k) |
|---|---|---|---|
| 0 | 3.8 | 4.5 | 1.69 |
| 30 | 4.5 | 4.7 | 1.78 |
| 60 | 0.4 | 3. | 1.82 |
| 90 | 2.8 | 7.2 | 1.7 |
| 120 | 8.3 | 8.8 | 1.97 |
| 150 | 7.5 | 8.2 | 2.49 |
| 180 | 9.9 | 8.4 | 2.72 |
| 210 | 14.8 | 9.5 | 2.7 |
| 240 | 14.3 | 9.2 | 2.88 |
| 270 | 17. | 9.9 | 3.34 |
| 300 | 12.6 | 10.3 | 2.84 |
| 330 | 4.1 | 6.7 | 2.23 |

**Table 4. Sector probability and Weibull shape and scale parameters for the Lillgrund site. Data obtained from the study of Göçmen and Giebel (2016)**