# Peer review of "Integrated wind farm layout and control optimization"

_Wind Energy Science, 2020_

## Referee Comment (RC1) · Anonymous Referee #1 · 17 Apr 2020

**Review of wes-2020-31**

**Overview**

The manuscript, "Integrated wind farm layout and control optimization" submitted by Mads M. Pedersen and Gunner Chr. Larsen offers an analysis on wind plant layout and control optimization, finding that the two procedures may be treated separately without significant reductions to the benefits to AEP. The work decoupling the numerical operations offers some real potential to engineering processes for wind plant design and operation. However, the results lack generality and insufficient detail is provided on the means by which results are attained. The manuscript would be greatly strengthened by a discussion of whether the control and layout optimization steps can always be safely decoupled. This would help to simplify wind plant optimization in general, which is an NP-hard problem. In summary, the manuscript offers some results that have a potential benefit to the wind energy research community and industry, but more information is required before results can be confidently and generally reproduced.

**Major Comments**

Wind plant control and operation by derating (i.e. axial induction control) is understood to be strongly dependent on the nature of the wake model used in wind plant performance modeling. No detail is provided on how wakes are modeled in the current work. Specifically, many well known velocity deficit and wake-added turbulence models exist and provide different estimates on the benefits of pitch-based derating.

The optimization procedure is discussed only briefly and will probably not be well understood by researchers that are not already familiar with the methods. Consequently, reproduction or verification of the results will be virtually impossible by other groups.

Finally, the layout optimization produces some results that do not seem to be appropriately constrained for implementation in reality, and may lead to overestimating the AEP gains. Turbines in the modeled wind plant appear to have been placed closer together than would be allowed in reality and in some cases are certainly operating in the near wake region of upstream turbines. Constraints or bounds on optimization parameters are not sufficiently clear in the manuscript.

Page 1: Authors state that the procedure outlined in the manuscript is, "the fastest possible optimization procedure". How is this determined? Can it actually be shown to be the fastest, or is it simply faster than a given alternative?

Page 1: Increasing AEP for any wind plant by 4% is a very substantial change. Is this estimate derived by comparing a modeled baseline production to a modeled controlled case, or is power production reported by the SCADA system used at all. There is no discussion of the effects of turbulence on wind turbine wakes or wind plant performance anywhere in the paper. This is a crucial consideration for wind turbine wake interaction and mixing and plays a huge role in the outputs of wake models used for power estimation. How is turbulence considered in the modeling, optimization, or estimation of AEP?

Page 5: "divergence free," isn't this another way of stating "conservation of mass" for incompressible flows?

Page 5: "linearity of the model, wakes from multiple upstream WTs can consistently be

superimposed to construct the flow field further downstream". How well do the authors expect this to reflect reality? From the governing equations, wake velocity deficits are highly non-linear. Other wake models use sum-of-squares superposition or maximum deficit approach. Why is a linear model assumed?

Equations (2) and (3): These relationships are derived through model outputs of HAWCStab2 by artificially limiting the power coefficient for a fixed wind speed. Am I correct in reading that there is no anlytical or empirical relationship to describe the modified values of Cp and Ct? Is this why the authors use look-up tables?

Page 6: "The results shown in Figure 1 can be used for the entire range of mean wind speeds requested for the system optimization," Cp and Ct are both functions of wind speed as well. What makes the authors confident that that the results in Fig. 1 are applicable at wind speeds other than 8 m/s, where they were defined? Equations (2) and (3) clearly state that Cp and Ct are functions of the mean wind speed, conditioned on tip-speed ratio and blade pitch angle.

Equation (5): Define limits of integration, wind direction is not typically discussed in units of radians. Is a resolution of 1 m/s for the numerical integration sufficient to capture rapid changes in region 2?

Figure 6, is the power production of each turbine relative to the nominal production at 8 m/s or is it relative to a different baseline value? Also, is the total power in Figure 7 just a sum of the plots in Figure 6?

Table 2: "Optimized Greedy 41.44 (+1.4%)" Results from 'sanity check' study indicate that optimal layout does not differ greatly from uniform spacing. Can the authors comment on the changes see in layout for case(2)? It also might be helpful to indicate computation time for each case, since that seems to be the justification for decoupling the layout and control optimizations.

Figure 13: Note case(1) on the left and case(3) on the right. Also case(3) results do not seem intuitively correct. There appear to be areas where a wind turbine could be placed within the central area of the wind plant that would reduce the need to derate to the same extreme of 19%. How closely spaced are the wind turbines on the western and southeastern edges? Given that Lillgrund is already a tightly packed wind plant, these may be dangerously close or impractical. Are any bounds provided for wind turbine spacing? Is the same optimal layout used for all wind directions, or is layout different for each case?

**Minor Comments**

Throughout the manuscript:
The naming convention of 'topology' is somewhat confusing. Suggest a change to WPP layout and WPP control, for clarity, as in the tables.
Commas and nested clauses are used with excess. For simplicity and readability, consider rephrasing with simpler, more direct messaging.
Phases are needlessly italicized throughout the text. Please remove text emphasis unless absolutely necessary.

Page 1: rephrase "in-stationary" as non-stationary or transient

Page 2: "A priory" -> a priori

Page 2: rephrase, "and if so then how"

Page 3:
rephrase as statement of research challenge rather than as a question.
"Is it possible to conduct WPP system optimization based on a full-blown CDF simulation of the complex WPP flow field with its complicated WT wakes interactions?"

Page 4: "wind direction and -speed" -> "wind direction and wind speed"

Page 4: rephrase, "that is possible for at the requested"

Page 5: Rephrase "but often is assumed uniform" as "but is often assumed to be uniform in practice" or similar.

Figure 2: cut in wind speed for a SWT2.3-93 is 3.5 m/s

Page 8: "i.e. the WT position coordinates", are the coordinates (lat, long) the two design variables or is there another?

Page 8: "Both of the above-sketched optimization", Only a single optimization strategy has been show so far.

Figure 4: caption should be on the same page as figure. Also consider making Figs 3 and 4 subfigures.

Use the \citet{} command for textual citations throughout the manuscript.

Page 9: How is the global optimum identified? In other words, how are the authors sure that the solution represents a global solution rather than a local optimum?

Table 1, insert comma after "layout"

Page 10: "both WT1 and WT2 is" -> "both WT1 and WT2 are"

Page 11: "the Cp- and the Ct dependence" -> "the dependence of Cp and the Ct on wind speed"

Figures 8 and 9: update vertical axis label. Simply writing "%" does not indicate what relative value is being considered.

Page 14: considerable -> considerably

Page 14: only insignificantly -> not significantly

Table 3: "4 Optimized Optimized(nested) -", remove from table if not pursued in analysis.
Caption should be on same page as table

Page 15: analogue -> analogous

Page 15 "more than doubled compared to the" -> "more than double that of"

Page 17: "Introductory," seems out of place.

---

## Short Comment (SC1) · 30 Apr 2020

Despite of successful field testing results of wake deflection (such as https://www.nrel.gov/docs/fy19osti/73991.pdf or https://www.wind-energ-sci.net/2/229/2017/), Pedersen and Larsen seem to come to the conclusion in their introduction that wake deflection is of smaller importance compared to wind turbine derating/axial induction based control. They refer to presentation slides from Andersen (2019) with LES studies, in addition to comparisons using low-fidelity models by Deshmukh. It seems to me however that in the slides by Andersen (2019), also the power of the combination of turbines cannot be increased through turbine derating, see slide 13/16?

The authors are referring to Gebraad, 2015 and mention that it is not taking into account the combination of pitch and TSR, thereby referring to Larsen's own work (Vitulli, 2019) where this was done. The combination of pitch and TSR is interesting, but it is questionable whether the linearized RANS modeling tool (FUGA) that was used by Vitulli can be used to compare results. The main points following out of Gebraad, 2015 and related publication (J. Annoni et al. Analysis of axial-induction-based wind plant control using an engineering and a high-order wind plant model. Wind Energy, 2015.), which was based on high-fidelity LES simulations using actuator-line rotors, are that when reducing thrust, also wake recovery is reduced because of reduced turbulence in the wake, limiting the potential of axial-induction based wake control. Secondly, when using pitch control in particular, the energy that is conserved in the wake is concentrated at the edge of the wake, so that most of that energy cannot be recovered at the downstream turbine. The linearized RANS modeling tool with actuator discs used in Vitulli's (and also now in Pedersen's work) might not be able to capture such effects?

This is not to say that there could not be benefits of axial induction control, but it might be only applicable in very tightly spaced wind farms, and probably smaller than expected by the model used in Pedersen and Larsen. A recent paper (Effects of axial induction control on wind farm energy production-A field test, van der Hoek, 2019) shows benefits of axial induction control to be present, but smaller than expected from a CFD model (FarmFlow). The row production increase in below-rated conditions is reported to be 3.3%, while the spacing is more tight than the Lillgrund row where 8% increase in AEP was predicted by Pedersen and Larsen. Perhaps by recreating the scenarios, Pedersen and Larsen could have a critical look at their model's predictions compared to field testing results.

---

## Short Comment (SC2) · 11 May 2020

Yes, it seems that the combination of RPM and pitch is still interesting to still consider in axial-induction based wake control (perhaps mostly for closely spaced wind farms), as was discussed in earlier work "Optimal open loop wind farm control" by Vitulli.

I would still like to add some critical notes to your reply:

- Deshmukh and Alison used an early version of the FLORIS Multizone wake model with a FLORIDyn extension (aimed at modeling wake steering) for optimizing axial induction based control. This version of FLORIS/FLORIDyn Multizone has been shown to be inaccurate for optimizing axial-induction based control. Extensions to the FLORIS Multizone model are made in Annoni et al (DOI: 10.1002/we.1891) to match LES results with axial-induction based control, in which case the predicted benefit of axial induction based control disappears (at least if we use pitch control separately).

- In your reply you state that Fuga is "not able to capture all aspects of wake flow and WT interaction", while in the paper, the model is advertised as a "full-blown CDF (sic.) simulation of the complex WPP flow field with its complicated WT wake interactions". There seems to be a mismatch in formulation. I think it would also be good to refer to the state-of-the-art in this field of research, where wake controls optimizations are in fact done with LES code (for example https://doi.org/10.3390/en11010177).

---

## Author Comment (AC1) · 11 May 2020

**Author's reply to 'Review of wes-2020-31' by Anonymous Referee #1**

Thank you very much for your detailed and valuable review of our manuscript. Below you find a copy of the referee's comments together with our responses marked in red.

**Review of wes-2020-31**

**Overview**

The manuscript, "Integrated wind farm layout and control optimization" submitted by Mads M. Pedersen and Gunner Chr. Larsen offers an analysis on wind plant layout and control optimization, finding that the two procedures may be treated separately without significant reductions to the benefits to AEP. The work decoupling the numerical operations offers some real potential to engineering processes for wind plant design and operation. However, the results lack generality and insufficient detail is provided on the means by which results are attained. The manuscript would be greatly strengthened by a discussion of whether the control and layout optimization steps can always be safely decoupled. This would help to simplify wind plant optimization in general, which is an NP-hard problem. In summary, the manuscript offers some results that have a potential benefit to the wind energy research community and industry, but more information is required before results can be confidently and generally reproduced.

It is correct, that it would greatly strengthen the generality of the manuscript if we could 'prove' that control and layout optimization can always be safely decoupled. This, however, implies that all possible generic integrated topology and wind farm control problems have this property - i.e. an arbitrary number of WTs, related arbitrary area constraints (i.e. WT 'density'), area shapes, wind climates etc. This is, unfortunately, not possible for this complex optimization problem, where an analytical solution providing such results are not possible.
This is also stated in the paper referring to the conclusion of Fathy et al., (2001): "A priory, it is not possible to evaluate whether the coupling between system design variables and system control variables is weak or strong for a complicated physical system like a WPP".

**Major Comments**

Wind plant control and operation by derating (i.e. axial induction control) is understood to be strongly dependent on the nature of the wake model used in wind plant performance modeling. No detail is provided on how wakes are modeled in the current work. Specifically, many well-known velocity deficit and wake-added turbulence models exist and provide different estimates on the benefits of pitch-based derating.

Rotors are modeled as actuator discs in a linear CFD framework - which in the paper is formulated as "The WTs are modelled as *actuator discs*, which in general can be vertically inhomogeneous, but often is assumed uniform". We will add that this model is used to model the wakes.
We are not modelling wake-added turbulence, as we are not considering WT loading but only WT production.

The optimization procedure is discussed only briefly and will probably not be well understood by researchers that are not already familiar with the methods. Consequently, reproduction or verification of the results will be virtually impossible by other groups.

We have added more details on the optimization procedure in the revised paper.

Finally, the layout optimization produces some results that do not seem to be appropriately constrained for implementation in reality, and may lead to overestimating the AEP gains. Turbines in the modeled wind plant appear to have been placed closer together than would be allowed in reality and in some cases are certainly operating in the near wake region of upstream turbines. Constraints or bounds on optimization parameters are not sufficiently clear in the manuscript.

The optimization is constrained by a minimum allowable distance to the nearest neighboring WT of 2D together with the convex boundary around the initial WPP layout. We will write this explicitly in section 2.4.

Page 1: Authors state that the procedure outlined in the manuscript is, "the fastest possible optimization procedure". How is this determined? Can it actually be shown to be the fastest, or is it simply faster than a given alternative?

The point is that de-composed nested approach is much faster that the fully integrated optimization approach. We will reformulate this phrase.

Page 1: Increasing AEP for any wind plant by 4% is a very substantial change. Is this estimate derived by comparing a modeled baseline production to a modeled controlled case, or is power production reported by the SCADA system used at all. There is no discussion of the effects of turbulence on wind turbine wakes or wind plant performance anywhere in the paper. This is a crucial consideration for wind turbine wake interaction and mixing and plays a huge role in the outputs of wake models used for power estimation. How is turbulence considered in the modeling, optimization, or estimation of AEP?

Yes, 4% is a substantial change, but of the same order of magnitude as reported in other papers using different approaches. This result, however, is based on the assumption that the inflow to the wind farm is homogeneous, stationary and well-known. Furthermore, controller-specific constraints such as tower-exclusion zone and smooth transition between regions are not considered. We will state this in the manuscript.

The 4% increase is relative to the model base line to obtain a consistent comparison. Comparing to SCADA data production would not make sense as model uncertainty then enter the 'equation'.

The effect of the ambient mean wind shear and turbulence characteristics on the wakes are specified in terms of a terrain roughness height, which in turn implicitly dictates the ambient turbulence conditions via the turbulence closure of the CFD model. We have included the applied roughness height and the turbulence intensity it dictates in the revised manuscript.

Page 5: "divergence free," isn't this another way of stating "conservation of mass" for incompressible flows?

Yes. We will add this.

Page 5: "linearity of the model, wakes from multiple upstream WTs can consistently be superimposed to construct the flow field further downstream". How well do the authors expect this to reflect reality? From the governing equations, wake velocity deficits are highly non-linear. Other wake models use sum-of-squares superposition or maximum deficit approach. Why is a linear model assumed?

We use the linear sum because it is consistent with the linear perturbation expansion of the Navier-Stokes equations in Fuga. We expect it to be at least as good as conventional empirical engineering models for wake summation. Full-scale validation studies of Fuga shows good agreement between model predictions and reality. We will add references to these studies

Equations (2) and (3): These relationships are derived through model outputs of HAWCStab2 by artificially limiting the power coefficient for a fixed wind speed. Am I correct in reading that there is no anlytical or empirical relationship to describe the modified values of Cp and Ct? Is this why the authors use look-up tables?

Yes, it is correct. Based on outputs from HAWCStab2 (numerical simulations of the rotor aerodynamics based on rotor/blade aerodynamic characteristics and using the BEM (Blade Element Momentum) approach) for a range of rotor speeds and pitch angles, the relationship between Cp and Ct is found by finding the rotor speed and pitch angle that, for each value of Cp, result in the lowest possible Ct. This relation is stored in look-up tables.

Page 6: "The results shown in Figure 1 can be used for the entire range of mean wind speeds requested for the system optimization," Cp and Ct are both functions of wind speed as well. What makes the authors confident that that the results in Fig. 1 are applicable at wind speeds other than 8 m/s, where they were defined? Equations (2) and (3) clearly state that Cp and Ct are functions of the mean wind speed, conditioned on tip-speed ratio and blade pitch angle.

Thrust scales with $U^2$ and the thrust coefficient is normalized with $U^2$. Power scales with $U^3$ and the power coefficient is normalized with $U^3$. These relations, however, assumes that the deformation of the blades are not wind-speed dependent. In reality the static blade deformation depends on the wind speed – that is why Ct and Cp depends on U in equations (2) and (3). In this study we simplify the model by using the static rotor deformation corresponding to 8 m/s for all wind speeds, and we do not expect the optimization result to be significantly affected by this simplification. We will add this information to the manuscript. For a more general formulation (also mentioned in the paper as a potential possibility when describing the rotor aerodynamics using HAWC2Stab), where the dependence of the static rotor/tower deflection with mean wind speed is taken into account, the Ct and Cp results will have to be computed for each relevant mean wind speed.

Equation (5): Define limits of integration, wind direction is not typically discussed in units of radians. Is a resolution of 1 m/s for the numerical integration sufficient to capture rapid changes in region 2?

We will replace radians with degrees.
The plot below shows the change in AEP when increasing the wind speed resolution from 0.1 m/s to 2 m/s. It is seen that the "error" in AEP for a wind speed resolution of 1 m/s is less than 0.15%. We consider this accuracy sufficient, and we expect the error to be much lower when comparing two cases,

which are both calculated with a resolution of 1 m/s.

[Figure]

Figure 6, is the power production of each turbine relative to the nominal production at 8 m/s or is it relative to a different baseline value? Also, is the total power in Figure 7 just a sum of the plots in Figure 6?

Yes, both are correct, except that the power production refers to a mean wind speed equal to 10 m/s. We will clarify this in the manuscript.

Table 2: "Optimized Greedy 41.44 (+1.4%)" Results from 'sanity check' study indicate that optimal layout does not differ greatly from uniform spacing. Can the authors comment on the changes see in layout for case (2)? It also might be helpful to indicate computation time for each case, since that seems to be the justification for decoupling the layout and control optimizations.

The degree of freedom of the "sanity check"-example is limited to movement of the middle turbine. The result depends on the trends of of the power and thrust curves as well as the wind speed distribution. With the input used in this study, the optimal position is very close to the initial layout (equal spacing), and therefore only an infinitesimal increase is optained.

In the 8-WT example, the optimizer has the freedom to move all turbines, and in this case the optimal layout is different from the initial layout. The plot below shows the cumulated AEP of the 8 WTs relative to the base line (i.e. initial layout and greedy control) as well as the position of the WTs (indicated by the dot markers). The green curve shows the optimal layout with greedy control (Case 2). It is seen that the distance between the two most upstream and the tree most downstream WTs are smaller than in the base case. This allows larger spacing and thereby production of the middle turbines, which, in this case, results in an increase of the AEP of the whole row. Obviuosly, this strategy is not possible with only three turbines.

[Figure]

The computation time is indeed an argument for the decoupling. Otherwise, the problem solution for a WF of the Lillgrund size becomes a major computational challenge - even on a big cluster. The computation times for the 8-WT row are listed below and will be added to table 2 of the paper.

| Case | Layout | Control | AEP [Gwh] | CPU time* [s] |
|---|---|---|---|---|
| 0 | Initial | Greedy | 40.85 | 0.01 |
| 1 | Initial | Optimized | 44.10 (+8.0%) | 4.20 |
| 2 | Optimized | Greedy | 41.44 (+1.4%) | 2.84 |
| 3 | Optimized | Optimized (sequential) | 44.558 (+9.1%) | 6.92 |
| 4 | Optimized | Optimized (nested) | 44.560 (+9.1%) | 3731 |

* On standard laptop PC

Figure 13: Note case(1) on the left and case(3) on the right. Also case(3) results do not seem intuitively correct. There appear to be areas where a wind turbine could be placed within the central area of the wind plant that would reduce the need to derate to the same extreme of 19%. How closely spaced are the wind turbines on the western and southeastern edges? Given that Lillgrund is already a tightly packed wind plant, these may be dangerously close or impractical. Are any bounds provided for wind turbine spacing? Is the same optimal layout used for all wind directions, or is layout different for each case?

Yes, we are using the same layout for all wind directions (as you cannot move WT when the wind direction changes), while the WT control settings varies with the wind direction.
The layout shown in Figure 13 is found to result in the highest overall AEP given the site conditions (wind speed distributions and wind direction frequency) used in this study. The inflow situation in Figure 13 reflects 10m/s from 223 deg, only, and for this particular flow case, the layout is obviously not optimal. The optimizer places the WT very close on the western and south-eastern edges, but does not violate the 2D spacing constraint (indicated by the dashed circles)

**Minor Comments**

Throughout the manuscript:

Thank you very much for these comments. We will replace and rephrase as suggested.

The naming convention of 'topology' is somewhat confusing. Suggest a change to WPP layout and WPP control, for clarity, as in the tables.

Commas and nested clauses are used with excess. For simplicity and readability, consider rephrasing with simpler, more direct messaging.

Phases are needlessly italicized throughout the text. Please remove text emphasis unless absolutely necessary.

Page 1: rephrase "in-stationary" as non-stationary or transient

Page 2: "A priory" -> a priori Page 2: rephrase, "and if so then how"

Page 3: rephrase as statement of research challenge rather than as a question. "Is it possible to conduct WPP system optimization based on a full-blown CDF simulation of the complex WPP flow field with its complicated WT wakes interactions?"

Page 4: "wind direction and -speed" -> "wind direction and wind speed"

Page 4: rephrase, "that is possible for at the requested"

Page 5: Rephrase "but often is assumed uniform" as "but is often assumed to be uniform in practice" or similar.

Figure 2: cut in wind speed for a SWT2.3-93 is 3.5 m/s

According to Siemens, there is no well-defined cut-in wind speed for the wind turbine used in the paper. In this study, we calculate the AEP based on power calculations for the wind speeds, 3, 4, ... , 25. In any case, the impact of the lowest wind speeds on the AEP are minimal and, in our opinion, insignificant for a layout study.

Page 8: "i.e. the WT position coordinates", are the coordinates (lat, long) the two design variables or is there another?

Yes, the two WT position design variables are the Cartesian x and y coordinates of the WT.

Page 8: "Both of the above-sketched optimization", Only a single optimization strategy has been show so far.

In the first strategy, all N(2+Nd Ns) design variables are optimized in one process, which is infeasible within the current frame work. The second strategy is the two-step nested approach. We will try to make this more clear in the manuscript.

Figure 4: caption should be on the same page as figure. Also consider making Figs 3 and 4 subfigures.

Use the \citet{} command for textual citations throughout the manuscript.

Page 9: How is the global optimum identified? In other words, how are the authors sure that the solution represents a global solution rather than a local optimum?

We cannot be sure that the presented solution is a global optimum, but we are confident that the result is close to optimal as we ran several long-run instances of the random search without getting a better result.

Table 1, insert comma after "layout"

Page 10: "both WT1 and WT2 is" -> "both WT1 and WT2 are"

Page 11: "the Cp- and the Ct dependence" -> "the dependence of Cp and the Ct on wind speed"

Figures 8 and 9: update vertical axis label. Simply writing "%" does not indicate what relative value is being considered.

Page 14: considerable -> considerably

Page 14: only insignificantly -> not significantly

Table 3: "4 Optimized Optimized(nested) -", remove from table if not pursued in analysis. Caption should be on same page as table

Page 15: analogue -> analogous

Page 15 "more than doubled compared to the" -> "more than double that of"

Page 17: "Introductory," seems out of place.

---

## Author Comment (AC2) · 11 May 2020

Dear Pieter Gebraad

Thank you for your interest in our study and for your comments.

We are not intentionally concluding that wake deflection is of smaller importance compared to wind turbine derating/axial induction based control:

We are referring to Deshmukh and Allison (2017) who obtains 0.9% increase in AEP for wake deflection, 6.6% for wake expansion and propagation (axial induction control) and 17.7% when combining the two strategies.

We are referring to Andersen (2019) who based on his two-turbine investigation finds that for a given reduction of the upstream WT thrust, the yaw wake deflection strategy is penalized more severely than the derating strategy measured in terms of aggregated power production of the two turbines analyzed in the study case. We will rephrase the sentence "The same conclusion was reached by Andersen (2019)" as this is not the case.

We are, however, arguing that axial induction control should not be rejected insignificant based on studies that uses either pitch regulation or rotor-speed regulation isolated, as a combination of pitch and rotor-speed regulation in most cases results in more power for the same level of thrust.
The plot below shows the thrust curves obtained with combined pitch and rotor-speed regulation (dashed lines) and with pitch regulation only (dotted lines). It is clearly seen that the combined pitch and rotor-speed regulation results in lower thrust for the same power production.

[Figure]

It is correct that slide 13 in Andersen (2019) does not show significant power gains for derating. According to the author, however, the derating strategy applied in this study is based on pitch regulation only. We therefore believe that this study do not reveal the full potential of derating.

It is also correct that Fuga, as well as all other models, are not able to capture all aspects of wake flow and WT interaction. In this case, however, simplifications of the flow field modeling is inescapable. It is simply not realistic to do wind farm layout and control optimization using a full non-linear LES coupled to meso-scale models for correct flow boundary conditions. Therefore, the question is how to simplify in the most adequate way. An alternative, widely used, simplification is to describe the wind-farm flow

field by superposition of engineering/empirical single wake models. However, we consider the present direct solution to the wake affected wind farm flow field as an innovative, valid and competitive alternative to the traditional single-wake-based approach as it provides a consistent solution to the full set of linearized Navier-Stokes equations and thus avoids the challenging inconsistent merging of engineering single wakes into a wind farm flow field.

The mentioned recent paper (Effects of axial induction control on wind farm energy production-A field test, van der Hoek, 2019) is indeed interesting, and we will refer to it in our manuscript. Their FarmFlow simulations shows an increase of 5.6 %, which we consider in the same order of magnitude as our 8%. The numbers are, however, not directly comparable due to different turbine spacing, wind speed distributions, inclusion/exclusion of above-rated wind speeds and derating strategies (van der Hoek (2019) derates using pitch regulation only).

In the numerical study by van der Hoek (2019) as well as ours, the inflow field is assumed homogenous, stationary and well known. Furthermore, controller-technical and practical details such as tower exclusion zones and smooth transition between regions not are considered. In the field test by van der Hoek (2019), for instance, the derating is applied via a two-level pitch offset as the optimal pitch setting was too complicated to implement in the controller.
The power increase of 3.3% seen in the field test is therefore, in our opinion, surprisingly high compared to the simulation results. In any case, it confirms that the potential of axial induction control is worth to investigate.

---

## Author Comment (AC3) · 15 May 2020

Dear Pieter Gebraad

Thank you for your follow up comments - please, find our comments to those below in red.

*-Yes, it seems that the combination of RPM and pitch is still interesting to still consider in axial-induction based wake control (perhaps mostly for closely spaced wind farms), as was discussed in earlier work "Optimal open loop wind farm control" by Vitulli.*

We assert that for a *given* Cp contour, the intelligent choice of Ct is the smallest possible, because it results in the smallest/weakest possible wake. This is justified in our earlier work "Optimal open loop wind farm control", and can in general only be obtained using two design variables (pitch and rotational speed). The superiority of this strategy is in our opinion independent/unaffected of/by WT spacing - the weakest possible wake for fixed production is, in our opinion, always preferable from a wind farm production perspective.

I would still like to add some critical notes to your reply:

- Deshmukh and Alison used an early version of the FLORIS Multizone wake model with a FLORIDyn extension (aimed at modeling wake steering) for optimizing axial induction based control. This version of FLORIS/FLORIDyn Multizone has been shown to be inaccurate for optimizing axial-induction based control.

Thank you for the information about the inaccuracy of the FLORIS/FLORIDyn Multizone version used by Deshmukh and Alison (2017). This is of course unfortunate, but their paper has not been redrawn or updated, and we have not found public work that states that their conclusions are invalid, or that they are using an inaccurate FLORIS/FLORIDyn Multizone version. In fact, they do not mention FLORIS nor FLORIDyn in their paper. On this basis, we cannot dismiss their work as it is the only other study that combines layout and derating optimization as far as we know. We will, however, keep it in mind and not draw conclusions based on their work alone.

- Extensions to the FLORIS Multizone model are made in Annoni et al (DOI: 10.1002/we.1891) to match LES results with axial-induction based control, in which case the predicted benefit of axial induction based control disappears (at least if we use pitch control separately).

We had a look into the mentioned publication by Annoni et al (DOI: 10.1002/we.1891). With the 3.3% power increase, seen in the field study by van der Hoek (2019), in mind we find it strange that Annoni et al. finds no benefit of axial-induction-based control. The suggested empirical ad hoc modification of the basic Jensen top-hat model for turbines on a single row (fully overlapping wakes) is based on a fitting to SOWFA (CDF LES) simulations for one single ambient inflow speed (8m/s) and one ambient turbulence intensity (6%). For the investigated cases, the SOWFA simulations shows very little effect of the investigated de-rating cases. However, these are, as you correctly mention, obtained by adjusting the (collective) pitch setting only, which is sub-optimal (cf. the above comment on an optimal WT de-rating setting). The FLORIS-fitting is based on an adjustment of the Jensen expansion rate, which is argued to relate to the upstream WT axial induction parameter. It is difficult for us to follow the physical reasoning behind this relationship. We consider the downstream linear wake expansion assumption in the Jensen model to be primary dictated by wake meandering (cf. E. Machefaux et al. (2014). Empirical Modeling of Single Wake Advection and Expansion using Full Scale Pulsed Lidar based Measurements; WE) and consequently associated not only with turbulence intensity but also with turbulence structure (primary

turbulence length scale). The paper by Annoni et al. clearly illustrates, that model simplifications comes with a price - more to this in the following.

- In your reply you state that Fuga is "not able to capture all aspects of wake flow and WT interaction", while in the paper, the model is advertised as a "full-blown CDF (sic.) simulation of the complex WPP flow field with its complicated WT wake interactions". There seems to be a mismatch in formulation. I think it would also be good to refer to the state-of-the-art in this field of research, where wake controls optimizations are in fact done with LES code (for example https://doi.org/10.3390/en11010177).

Perhaps this formulation about the capabilities of Fuga is unclear. What is really meant is that no model - including Fuga - is able to capture all aspects of wake flow and WT interaction. CFD LES is also an approximate approach, where the smaller scales are modelled, and where the precise interface to mesoscale flow boundary conditions is still an open question. Regarding the actuator line approach often used together with CFD LES simulations (also in SOWFA), work is still ongoing regarding the regularization kernel convolution of the blade forces (to avoid singularities in the numerical formulation), which also has an impact on WT power production (cf. recent work on Actuator-Line-Smearing-Correction by Alexander Meyer Forsting, Georg Pirrung and Néstor Ramos García; Wind Energ. Sci., 4, 369–383, 2019). Thus, high fidelity CFD simulation is not an unambiguous concept … and not necessarily the universal truth.

Fuga is a "full-blown CDF solver" in the sense that the solver consistently solves a set of first principles NS equations. This set is simplified as well as conventional CFD RANS and CFD LES are simplified, but it remains a "full-blown CDF solver".

---

## Referee Comment (RC2) · Anonymous Referee #2 · 28 May 2020

**wes-2020-31**

**Overview**

"Integrated wind farm layout and control optimization", written by Mads Pedersen and Gunner Larsen is generally well organized and well written. A wind farm layout and control optimization methodology is presented and analyzed. An approach to separating the control and layout optimizations is presented that is seemingly useful, but could use some further validation. The problem formulation is generally well presented, but could benefit from more detailed descriptions to make it easier for researchers to reproduce the results. There are a number of typographical and grammatical corrections that would improve the quality of the work, though none of them are extremely drastic. Finally, more discussion of the flow-field assumptions made may significantly reduce any doubt of the methods from the community.

**High-level Comments**

The use of two "example" studies to provide some intuition on the optimizing functions is a good way to tell the story, but the second example could provide more. By considering a single row of WTs, you are effectively removing a degree of freedom for the optimization solver to handle. It would be nice to see whether the AEP differences between the sequential and nested optimizations are still small for a minimal working example that doesn't remove design variables, but just has a few turbines and an appropriately constrained space. This would help to remove any doubt that the sequential approach is sound.

There is no mention or discussion of the wake model. This would be good to know more about, given that the proposed control method is axial induction control. Additionally, there is brief mention of the existence of wind shear and turbulence – details on this would be useful.

Especially because the optimized Lillgrund WPP has turbines that are so close together, some discussion of the validity of the linear CFD solver's ability to accurately represent the near wake region of upstream turbines would be very helpful. This goes hand in hand with the need for a discussion on the wake model.

In the very beginning you note that "The purpose of this paper is to investigate the influence of optimal wind farm control on the wind farm layout". To me, the word *influence* feels misleading. If I am not mistaken, this work really is investigating the joint optimization of the layout and control, not specifically the *influence* of one on the other. It may be worth it to consider re-phrasing this, or restructuring the paper a bit make it clear that the influence is in fact being investigated.

You *briefly* mention that the only location constraint is a minimum of 2D from the nearest WT and the wind farm boundaries. What motivated this distance? This (and any other constraints that exist) might better fit in the problem formulation, not in the conclusion.

The introduction offers a fairly good review of the relevant literature but could benefit from some revision and restructuring. There are a lot of sentences that are extended through a series of commas, semicolons, and dashes and can feel tedious.

The presentation of the use of a "detailed aero-servo-elastic model" for the optimization approach (P4.L26) is a bit of a stretch. It seems that the optimization itself uses simplified models (so-called "surrogates"), that are rooted in steady-state BEM solvers, but complete aero-elastic analysis is not done for the optimization. There certainly does not seem to be any dynamic "servo models", just the assumption that the employed individual WT controller is capable of perfectly tracking the derated Cp/Ct.

**More detailed comments & formatting**

In my opinion, the use of italics to emphasize words is over-used. Sometimes it is useful, but caused a little confusion for me at times as well.

There are a lot of leading and trailing hyphens that are unnecessary throughout.

P1.L10 – Should be clear that you are focused on controlling the turbines for wind-farm wide AEP maximization, not just doing standard wind turbine control. "A priori" is unnecessary in this sentence.

P1.L20 – "…  a Swedish offshore wind farm …"

P1.L19 – "…the capability  of the developed …". Double check singular/plural adjectives throughout paper.

P1.L30 – "capital costs that depend on the WPP layout"

P2.L10 – "A priory" -> "A priori"

P3.L2 – modal or model?

P3.L10 – "tip speed ratio"

P3.L22 – I'm not sure I see how this work is specifically "guided by" the statement in 2). Confusingly, you state that , 2) suggested that more "realistic" studies introduce a lot more uncertainty, and then you say that you are attempting to get more "realistic" results. I am admittedly not very familiar with the work in (Kheirabadi and Nagamune, 2019), so perhaps I am missing something here.

P3.L23 – "CDF" -> "CFD"

P3.L32 – "CDF" -> "CFD"

P4.L16 – "justifies"

P5.L23 – "relies *on* an extended"

P7.L4 – "(3, 10, and 15 m s^-1 *are* marked). The marking of the rotor speed limits is probably unnecessary and clutters the figure.

P7.L17 – "this is assured to the highest possible degree" – what do you mean by this?

P7.L17 – "implementation *of* the shortcuts"

P8.L18 – what do you mean by "both of the above … will eventually lead to the same result"? Only one of the optimization approaches is sketched "above"

P8.L1 – ".. there *are* three *common* ways…"

P9.L10 – "A few local optima" is vague. This statement should be elaborated upon and/or justified more.

Figure 4 – make sure the figure caption is on the same page as the figure

Figure 5 – what do the black up/down arrows represent?

Figure 6 – Having the axis derating percentages range from 0%-100%, but the power percentage range from 40% - 100% in the colorbar is confusing

Figures 8 and 9 – The titles on these two figures should probably be the same, or similar. At least the y-axis label, and perhaps the titles, should reflect the fact that the percentage value plotted is percent of relative power.

P14.L7 – "considerablye"

P14.L11 – case (4), a the

(see major comments about my concerns with the lack of 2-D dimensionality in this second sanity check)

Table 3 – caption should be on same pages as table

P17.L10 – "Inherit" –> "inherent"

P17.L17 – "", This sentence is generally very confusing, and I am not sure I see the "clearly exaggeration"

---

## Author Comment (AC4) · 5 Jun 2020

**Author's reply to 'Review of wes-2020-31', Anonymous Referee #2**

Thank you very much for your detailed and valuable review of our manuscript. Below you find a copy of the referee's comments together with our responses marked in red.

**Overview**

"Integrated wind farm layout and control optimization", written by Mads Pedersen and Gunner Larsen is generally well organized and well written. A wind farm layout and control optimization methodology is presented and analyzed. An approach to separating the control and layout optimizations is presented that is seemingly useful, but could use some further validation. The problem formulation is generally well presented, but could benefit from more detailed descriptions to make it easier for researchers to reproduce the results. There are a number of typographical and grammatical corrections that would improve the quality of the work, though none of them are extremely drastic. Finally, more discussion of the flow-field assumptions made may significantly reduce any doubt of the methods from the community.

**High-level Comments**

The use of two "example" studies to provide some intuition on the optimizing functions is a good way to tell the story, but the second example could provide more. By considering a single row of WTs, you are effectively removing a degree of freedom for the optimization solver to handle. It would be nice to see whether the AEP differences between the sequential and nested optimizations are still small for a minimal working example that doesn't remove design variables, but just has a few turbines and an appropriately constrained space. This would help to remove any doubt that the sequential approach is sound.

We agree that a minimal example with two location degrees of freedom pr. WT would remove some doubt. To remove any doubt, however, the full example is required as the effects of both optimal layout and optimal derating is highly dependent on the number of turbines and the admissible site area and geometry. We have tried to optimize a minimal wind farm with 3x3 WTs, but the optimizer did not manage to find a trustworthy solution. We are therefore hesitating about adding such an example.

There is no mention or discussion of the wake model. This would be good to know more about, given that the proposed control method is axial induction control. Additionally, there is brief mention of the existence of wind shear and turbulence – details on this would be useful.

Rotors and wakes are modeled as actuator discs in a linear CFD framework - which in the paper is formulated as "The WTs are modelled as actuator discs, which in general can be vertically inhomogeneous, but often is assumed uniform". We will specify that this model is used to model the drag force, which in turn generates the wakes. The actuator disc's are modeled using the aerodynamic model described in detail in Section 2.2.

The effect of the ambient mean wind shear and turbulence characteristics on the wakes are specified in terms of a terrain roughness height, which in turn implicitly dictates the ambient turbulence conditions via the turbulence closure of the CFD model as based on Monin–Obukhov theory for neutral

atmospheric stratification, which is assumed. We have included the applied roughness height and the turbulence intensity it dictates at WT hub height in the revised manuscript.

Especially because the optimized Lillgrund WPP has turbines that are so close together, some discussion of the validity of the linear CFD solver's ability to accurately represent the near wake region of upstream turbines would be very helpful. This goes hand in hand with the need for a discussion on the wake model.

The linear CFD model does not rely on the assumption that the wind speed in the wake is U(1-2a) as e.g. the classical N.O. Jensen wake model. In principle, the wake model is therefore also valid in the near wake. The uniformly loaded actuator disc formulation, however, implies simplifications visible in the near wake region (no azimuthal or radial force variations etc.). Some distance downstream the effect from such variations vanishes, and we expect the model prediction to be acceptable 2D downstream.

In the very beginning you note that "The purpose of this paper is to investigate the influence of optimal wind farm control on the wind farm layout". To me, the word influence feels misleading. If I am not mistaken, this work really is investigating the joint optimization of the layout and control, not specifically the influence of one on the other. It may be worth it to consider re-phrasing this, or restructuring the paper a bit make it clear that the influence is in fact being investigated.

You are right. We will change this formulation.

You briefly mention that the only location constraint is a minimum of 2D from the nearest WT and the wind farm boundaries. What motivated this distance? This (and any other constraints that exist) might better fit in the problem formulation, not in the conclusion.

The minimum-spacing constraint is applied to avoid the turbines to be placed unrealistically close together. The minimum distance of 2D is chosen because it is around the minimum distance we have seen in a real wind farm.

We will list the applied constraints in the problem formulation where it belongs.

The introduction offers a fairly good review of the relevant literature but could benefit from some revision and restructuring. There are a lot of sentences that are extended through a series of commas, semicolons, and dashes and can feel tedious.

We will review the introduction and try to make it easier to read

The presentation of the use of a "detailed aero-servo-elastic model" for the optimization approach (P4.L26) is a bit of a stretch. It seems that the optimization itself uses simplified models (so-called "surrogates"), that are rooted in steady-state BEM solvers, but complete aero-elastic analysis is not done for the optimization. There certainly does not seem to be any dynamic "servo models", just the assumption that the employed individual WT controller is capable of perfectly tracking the derated Cp/Ct.

We are using the aero-servo-elastic tool HAWCStab2 with a detailed aero-elastic model of the turbine (i.e. no dynamic servos in the model) to establish a surrogate relationship between power and thrust. We will clarify this in the revised manuscript.

**More detailed comments & formatting**

Thank you very much for these comments, corrections and suggestions. We will replace and rephrase as suggested.

In my opinion, the use of italics to emphasize words is over-used. Sometimes it is useful, but caused a little confusion for me at times as well.

There are a lot of leading and trailing hyphens that are unnecessary throughout.

P1.L10 – Should be clear that you are focused on controlling the turbines for wind-farm wide AEP maximization, not just doing standard wind turbine control. "A priori" is unnecessary in this sentence.

P1.L20 – "… the a Swedish offshore wind farm …"

P1.L19 – "…the capability ies of the developed …". Double check singular/plural adjectives throughout paper.

P1.L30 – "capital costs that depends on the WPP layout"

P2.L10 – "A priory" -> "A priori"

P3.L2 – modal or model?

P3.L10 – "tip speed ration"

P3.L22 – I'm not sure I see how this work is specifically "guided by" the statement in 2). Confusingly, you state that , 2) suggested that more "realistic" studies introduce a lot more uncertainty, and then you say that you are attempting to get more "realistic" results. I am admittedly not very familiar with the work in (Kheirabadi and Nagamune, 2019), so perhaps I am missing something here.

P3.L23 – "CDF" -> "CFD"

P3.L32 – "CDF" -> "CFD"

P4.L16 – "justifies"

P5.L23 – "relies on an extended"

P7.L4 – "(3, 10, and 15 m s^-1 are marked). The marking of the rotor speed limits is probably unnecessary and clutters the figure.

P7.L17 – "this is assured to the highest possible degree" – what do you mean by this?

P7.L17 – "implementation of the shortcuts"

P8.L18 – what do you mean by "both of the above … will eventually lead to the same result"? Only one of the optimization approaches is sketched "above"

P8.L1 – ".. there are three common ways…"

P9.L10 – "A few local optima" is vague. This statement should be elaborated upon and/or justified more.

Figure 4 – make sure the figure caption is on the same page as the figure

Figure 5 – what do the black up/down arrows represent?

Figure 6 – Having the axis derating percentages range from 0%-100%, but the power percentage range from 40% - 100% in the colorbar is confusing

Figures 8 and 9 – The titles on these two figures should probably be the same, or similar. At least the y-axis label, and perhaps the titles, should reflect the fact that the percentage value plotted is percent of relative power.

P14.L7 – "considerablye"

P14.L11 – case (4), a the (see major comments about my concerns with the lack of 2-D dimensionality in this second sanity check)

Table 3 – caption should be on same pages as table

P17.L10 – "Inherit" –> "inherent"

P17.L17 – "Introductory", This sentence is generally very confusing, and I am not sure I see the "clearly exaggeration